# Bemcentinib as monotherapy and in combination with low-dose cytarabine in acute myeloid leukemia patients unfit for intensive chemotherapy: a phase 1b/2a trial

Beyond first line, the prognosis of relapsed/refractory (R/R) acute myeloid leukemia (AML) patients is poor with limited treatment options. Bemcentinib is an orally bioavailable, potent, highly selective inhibitor of AXL, a receptor tyrosine kinase associated with poor prognosis, chemotherapy resistance and decreased antitumor immune response. We report bemcentinib monotherapy and bemcentinib+low-dose cytarabine combination therapy arms from the completed BerGenBio-funded open-label Phase 1/2b trial NCT02488408 (www.clinicaltrials.gov), in patients unsuitable for intensive chemotherapy. The primary objective in the monotherapy arm was identification of maximum tolerated dose with secondary objectives to identify dose-limiting toxicities, safety and efficacy, and bemcentinib pharmacokinetic profile. In the combination arm, the primary objective was safety and tolerability, with efficacy and pharmacokinetics as secondary objectives. Safety and tolerability were based on standard clinical laboratory safety tests and Common Terminology Criteria for Adverse Events version 4. Bemcentinib monotherapy (32 R/R, 2 treatment-naïve AML and 2 myelodysplasia patients) was well-tolerated and a loading/maintenance dose of 400/200 mg was selected for combination treatment, comprising 30 R/R and 6 treatment-naïve AML patients. The most common grade 3/4 treatment-related adverse events were cytopenia, febrile neutropenia and asymptomatic QTcF prolongation, with no grade 5 events reported. In conclusion, bemcentinib+low-dose cytarabine was safe and well tolerated.

Acute myeloid leukemia (AML) represents an aggressive hematopoietic malignancy commonly affecting the elderly, with a median age of 68 years at diagnosis. The standard of care (SOC) as first line (1L) treatment is intensive chemotherapy (cytarabine and anthracyclines) followed by allogeneic stem cell transplantation, if appropriate[1].

However, patients > 65 years often respond poorly to induction chemotherapy, with a median survival < 1 year, as a result of unfavorable genomic features and increased treatment-resistance[2,3].

Moreover, elderly patients may not be eligible for intensive chemotherapy, due to age and/or comorbidities[4]. In this demographically growing population, less-intensive treatment regimens including hypomethylating agents (HMAs) (azacitidine or decitabine) and LDAC induce remissions (complete remission (CR) and CR with incomplete recovery (CRi)) in less than 35% of patients[5–8].

The addition of venetoclax to these less-intensive treatment regimens has resulted in increased response rates of 60−75% in the

e-mail: s.loges@dkfz-heidelberg.de

frontline setting, thus becoming the current SOC for patients ineligible for intensive therapy[9–12]. Most of these patients will relapse, and beyond 1 L the prognosis of older patients with relapsed or refractory AML (R/R-AML) is particularly poor[4]. The mOS in R/R-AML patients after failing 1 L therapy is 4 months, with only 2.9 months if the 1 L treatment was venetoclax-based and the patient received salvage therapy[13,14]. Response rate in R/R-AML is < 20% with HMAs or LDAC and 31% with HMA or LDAC plus venetoclax in venetoclax-naïve patients[15,16].

AXL, a member of the TAM (TYRO3, AXL, MERTK) receptor tyrosine kinase family, is overexpressed in AML and represents an independent predictor of poor OS[17–19]. AXL expression in AML is linked to pathobiology, and aberrant activation of AXL in tumor cells promotes proliferation, survival and therapy resistance[19–22]. AXL is also expressed on innate immune cells and promotes an immunosuppressive tumor microenvironment[23–27].

Bemcentinib is a first-in-class, potent, selective, oral inhibitor of AXL. Bemcentinib prevents AXL phosphorylation and downstream signaling in preclinical models of AML, resulting in reduced tumor growth, enhanced sensitivity towards therapy and enhanced antileukemic immune responses[19,25,28]. Here, we report safety and efficacy data for patients enrolled in a phase 1b/2a open label dose escalation and cohort expansion trial in AML and intermediate and high-risk MDS patients. We focus on the dose escalation cohorts treated with bemcentinib monotherapy (R/R-AML/high-risk MDS), and the cohorts treated with bemcentinib in combination with LDAC (R/R-AML and newly diagnosed AML). We also report pharmacokinetics (PK) of bemcentinib and correlate it to targeted inhibition of AXL and AXL downstream signaling.

## Results

### Dose escalation cohort
A total of 36 patients were enrolled in the dose escalation (DE) cohort of the study and administered bemcentinib as monotherapy. The primary endpoint for this cohort was identifying the maximum tolerated dose, with identification of the dose-limiting toxicity profile, exploration of safety and efficacy and characterization of bemcentinib pharmacokinetics as secondary objectives. All 36 patients were discontinued from the study prior to study completion; the reasons for discontinuation were progressive disease (PD) (19 patients, 52.8%); death (9 patients, 25.0%); adverse event (AE), investigator's decision, and subject withdrawal of consent (2 patients, 5.6% each); initiation of alternative cancer therapy and other (prolonged QTcF on ECG) (1 subject, 2.8% each) (Fig. 1).

Patients had relapsed or refractory AML/MDS following previous treatment with cytotoxic chemotherapy or a gene expression modulator (e.g. a demethylating agent). Disease characteristics are provided in Table 1. The median age in the DE cohort was 74.5 years (range 51–85 years) with a median ECOG performance status of 1 (range 0-2) at enrolment. 34 patients were diagnosed with AML (94%) and two with high-risk MDS.

### LDAC cohort
A total of 36 AML patients were enrolled in the phase 2a cohorts of bemcentinib + LDAC: B2 ($n = 16$) and B5 ($n = 20$) (Fig. 1). The primary endpoint for this cohort was the safety and tolerability of the combination, with exploration of efficacy and pharmacokinetics as secondary objectives. Patients and disease characteristics are provided in Table 1.

The median age was 76 years (range 66-86 years) with a median ECOG performance status of 1 (range 0–2) at enrolment. The AML cytogenetic profile was adverse in 14 (39%), intermediate in 13 (36%), favorable in 7 (19%) and not available for 2 (6%) patients. Of the 36 enrolled patients, 6 (17%) were treatment-naïve, 21 (58%) relapsed and 9 (25%) were refractory. Of the 30 R/R AML patients, 50% were in 2 L, and 50% in >2 L.

### Safety and tolerability
The phase 1b dose escalation was performed with two formulations of bemcentinib. Initially, formulation 1 was administered at loading (Days 1, 2)/maintenance doses of 400/100 mg ($n = 6$), 600/200 mg ($n = 14$) and 900/300 mg ($n = 5$). Subsequently, an advanced formulation was tested at loading (Days 1–3)/maintenance doses of 200/100 mg ($n = 4$) and 400/200 mg ($n = 6$). Three dose-limiting toxicities (DLT) were observed (Supplementary Data 1), although one occurred outside the DLT assessment window. Although the formal criteria for a maximum tolerated dose were not met (there was no dose in which two DLTs were observed within the first six patients), the 400/200 mg enhanced formulation dose was chosen as the recommended phase 2 dose for the LDAC cohort based on combined assessment of the DLTs and other adverse events.

The majority of treatment-emergent adverse events (TEAEs) in both the DE cohorts and the LDAC cohort were not considered to be related to study treatment and were mild or moderate in grade (Fig. 2, Table 2).

Fatal events, none of which were considered by the investigator to be related to study treatment, along with SAEs and TEAEs Grade ≥3 considered to be related to bemcentinib treatment are detailed in Supplementary Data 2 for both cohorts (combined $n = 72$). There were 7 fatal SAEs in the DE cohort and 4 in the LDAC cohort comprised of sepsis/neutropenic sepsis ($n = 4$), pneumonia/fungal pneumonia ($n = 3$), cerebral hemorrhage ($n = 2$), pulmonary oedema and multi-organ failure. None of the patients with fatal TEAEs had an objective response. The most common SAEs/TEAEs Grade ≥3 related to bemcentinib were anemia (DE $n = 1$, LDAC $n = 7$), QTcF prolongation (DE $n = 3$, LDAC $n = 4$), thrombocytopenia (DE $n = 3$, LDAC $n = 3$), neutropenia (DE $n = 2$, LDAC $n = 1$), and platelet count decreased (LDAC $n = 3$) (Supplementary Data 2).

The most commonly reported SAEs (reported in ≥10% patients in either cohort), and TEAEs of any grade (reported in ≥20% patients in either cohort), whether or not considered related to bemcentinib, are shown in Table 3. ECG QTcF prolongation was considered a TEAE of particular interest due to the potential severity and probable relationship to bemcentinib. Supplementary Data 3 summarizes all the observed ECG QTcF prolongation TEAEs by grade and details the most severe event per patient (grade, onset, duration and relationship to bemcentinib). Most patients experienced only Grade 1 or Grade 2 QTcF prolongation (DE $n = 5/9$, LDAC $n = 13/17$). Four DE patients and 4 LDAC patients experienced Grade 3 QTcF prolongation. No patients experienced Grade 4 or Grade 5 events.

TEAEs of grade ≥3 observed in ≥10% of patients are displayed in Fig. 2. A full list of TEAEs of grade ≥3 summarized by System Organ Class and Preferred term is provided in Supplementary Data 4. Only a single event of eye toxicity of grade ≥3 was observed, a secondary cataract that resolved without dose modification of bemcentinib.

4 patients (11%) in each of the DE cohort and LDAC cohort were discontinued from treatment due to TEAEs (Supplementary Data 5). 17 patients had dose interruptions due to adverse events, 3 patients had dose reductions, and 6 patients had bemcentinib withdrawn in the DE cohorts. 22 patients had dose interruptions, 4 patients had dose reductions, and 7 patients had bemcentinib withdrawn in the LDAC cohorts. Cytarabine was interrupted, dose-reduced and withdrawn in 8 patients, 4 patients and 1 patient, respectively (Supplementary Data 5). The median interval between cytarabine dosing cycles was 35 days (interquartile range 28–42).

### Treatment efficacy
Responses were assessed in the 36 patients treated with bemcentinib as monotherapy (DE cohort, Table 4). Overall, ORR was 14% (1 CR, 2 CRi, 1 Morphological Leukemia-Free State (MLFS) and one Marrow Response (MDS only)) but reached 29% in the 400/200 mg dosing group (1 CR, 1 CRi).

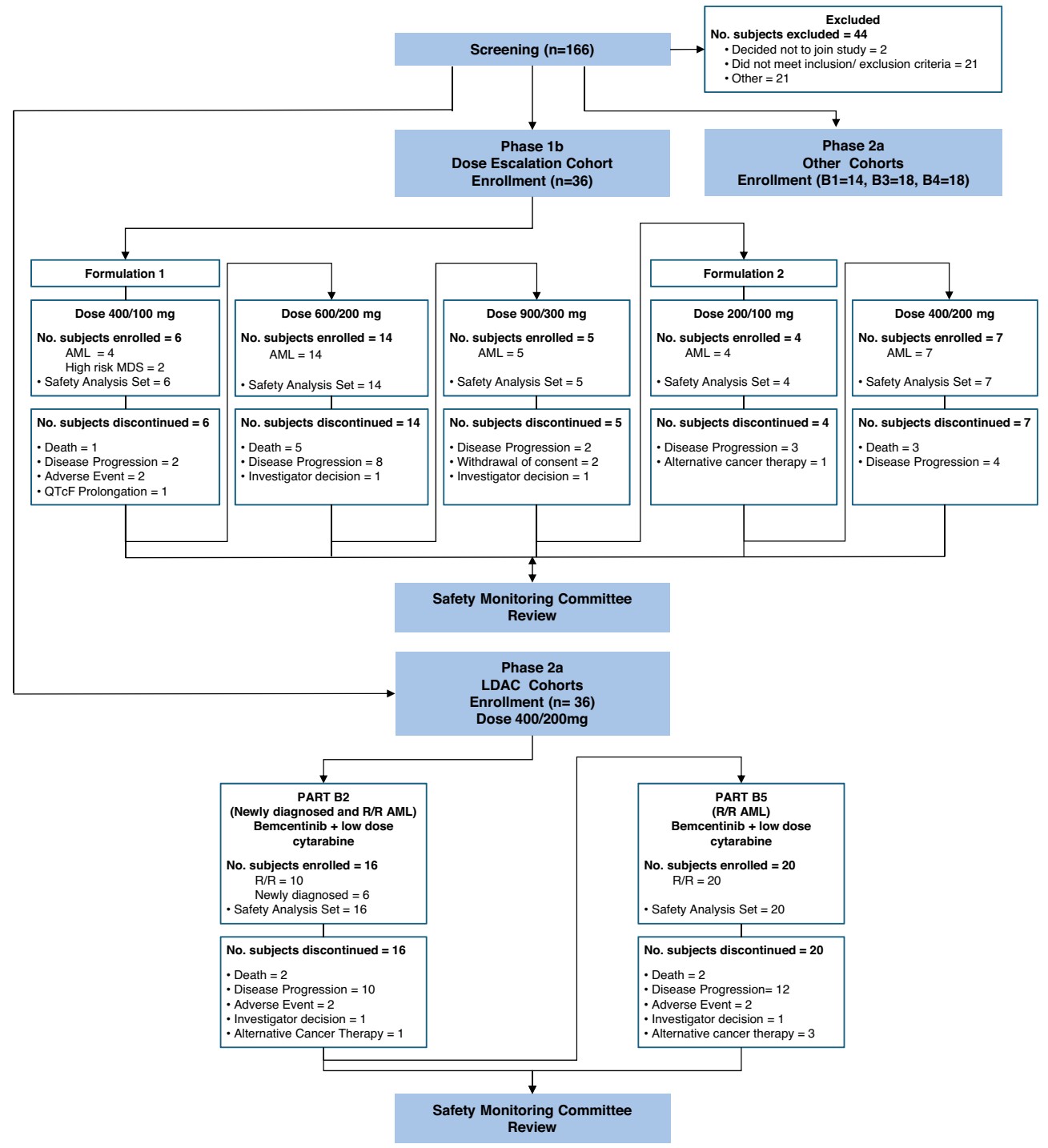

**Fig. 1 | Flowchart of patient disposition in the DE and LDAC cohorts.** AML, acute myeloid leukemia; LDAC, low-dose cytarabine; MDS, myelodysplastic syndrome; No., Number; R/R, relapsed/refractory.

In the patients treated with bemcentinib in combination with LDAC (LDAC cohort; Table 5), ORR was 50% (1 CR, 1 CRi and 1 partial remission [PR]) among the 6 treatment-naive patients and 20% (3 CR, 3 CRi) among the 30 R/R patients. DCR (including objective responses and patients with unchanged disease status for 3 treatment cycles) was 67% (4/6) for treatment-naive patients and 33% (10/30) for R/R patients.

Median time to objective response in the DE cohort was 22 days (95% CI 22–43); in the LDAC cohort, this number was 85 days (95% CI 85–163) for treatment-naive and 118 days (95% CI 64–371) for R/R patients.

The mOS (Fig. 3a) in the treatment-naïve patients ($n = 6$) was 16.1 months (95% CI, 1.8-NA). In the R/R ($n = 30$) patients, mOS was 7.8 months (3.9–11.2).

The median EFS (mEFS) was 16.5 months (1.8-NA) in treatment-naïve ($n = 3$) and 2.6 months (1.3–4.4) in R/R patients ($n = 30$) (Fig. 3b).

The median RFS (mRFS) in treatment-naïve ($n = 3$) compared to R/R ($n = 6$) patients was 17.3 months (13.7-NA) and 12.6 months (2.3-NA), respectively (Fig. 3c).

As only patients unfit for intensive chemotherapy were eligible for this trial, no patients were bridged to transplantation.

**Table 1 | Demographics and baseline characteristics for patients**

| Property | Phase 1b | | Phase 2a | |
|---|---|---|---|---|
| | Number of patients | Percentage, % | Number of patients | Percentage, % |
| **Sex [1]** | | | | |
| Female | 14 | 61 | 12 | 33 |
| Male | 22 | 39 | 24 | 67 |
| **Age (years)** | | | | |
| Median | 74.5 (Range 51–85) | | 76 (Range 66–86) | |
| <75 | 18 | 50 | 12 | 33 |
| ≥75 | 18 | 50 | 24 | 67 |
| **ECOG at screening** | | | | |
| Median | 1 (Range 0–2) | | 1 (Range 0–2) | |
| 0 | 7 | 19 | 13 | 36 |
| 1 | 19 | 53 | 18 | 50 |
| 2 | 10 | 28 | 5 | 14 |
| **Type of cancer** | | | | |
| AML | 34 | 94 | 36 | 100 |
| MDS (high risk) | 2 | 6 | 0 | 0 |
| **% blasts at screening (bone marrow)** | | | | |
| <10 | 4 | 11 | 5 | 14 |
| ≥10 | 26 | 72 | 31 | 86 |
| missing | 6 | 17 | 0 | 0 |
| **Cytogenetic Profile [2]** | | | | |
| favorable | 2 | 6 | 7 | 19 |
| intermediate | 3 | 8 | 13 | 36 |
| adverse | 7 | 19 | 14 | 39 |
| missing | 24 | 67 | 2 | 6 |
| **Disease status** | | | | |
| Treatment-naive | 2 | 6 | 6 | 17 |
| relapsed | 16 | 44 | 21 | 58 |
| refractory | 12 | 33 | 9 | 25 |
| other | 6 | 17 | 0 | 0 |
| **No. lines prior therapy** | | | | |
| Median | 2 (Range 1–6) | | 1 (Range 0–8) | |
| 0 | 1 | 3 | 6 | 17 |
| 1 | 12 | 33 | 15 | 42 |
| 2 | 11 | 31 | 9 | 25 |
| ≥3 | 12 | 33 | 6 | 17 |
| **Prior treatment** | | | | |
| Venetoclax | 0 | 0 | 11 | 31 |
| Allo-HCT | 2 | 6 | 0 | 0 |
| Intensive chemotherapy | 18 | 50 | 9 | 25 |
| **FLT3 status** | | | | |
| Wildtype | 17 | 47 | 21 | 58 |
| Mutated | 7 | 19 | 5 | 14 |
| of which ITD | 2 | 6 | 1 | 3 |
| Missing | 12 | 33 | 10 | 28 |

*Allo-HCT* Allogeneic hematopoietic cell transplantation, *AML* Acute Myeloid leukemia, *ECOG* Eastern Cooperative Oncology Group performance status, *ITD* internal tandem duplication, *MDS* Myelodysplastic syndrome, *No.* number. Headings and subheadings in bold.
[1] Self-reported.
[2] Cytogenetic risk was assessed by each study site using locally defined criteria. DE cohort patients were assessed as low, medium or high risk. LDAC Cohort patients were assessed as favorable, intermediate or adverse risk. Where specific data was provided, classification was consistent with ELN 2017 criteria.

## Pharmacokinetic assessment

PK data for the DE cohort was available for a total of 10 patients receiving a maintenance dose of 100 mg bemcentinib daily, 21 patients receiving 200 mg daily and 5 patients receiving 300 mg daily. The mean $C_{max}$ at steady state was $140 \pm 110$ ng/ml, $162 \pm 121$ ng/ml and $398 \pm 200$ ng/ml for the maintenance doses 100 mg, 200 mg and 300 mg respectively. The mean $C_{trough}$ at steady state was $136 \pm 109$ ng/mL, $154 \pm 118$ ng/mL and $382 \pm 193$ ng/mL, and AUC was $3310 \pm 2630$ ng•h/mL, $3810 \pm 2860$ ng•h/mL and $9390 \pm 4720$ ng•h/mL. The median half-life ($t_{1/2}$) of bemcentinib at each dose level was 159 (95% CI 124–310), 124 (95% CI 84–199) and 156 (95% CI 87-384) hours, respectively.

PK data was available for a total of 35 patients in the LDAC cohort following administration of bemcentinib in combination with LDAC. The mean $C_{max}$ at steady state following a maintenance dose of 200 mg of bemcentinib daily was $210 \pm 128$ ng/ml for the LDAC cohort. The mean $C_{trough}$ at steady state was $196 \pm 124$ ng/ml and mean AUC was $4890 \pm 3020$ ng•h/mL. The median $t_{1/2}$ of bemcentinib in LDAC cohort patients was 124 (95% CI 97–142) hours.

## Pharmacodynamic assessments

Assessment of potential pharmacodynamic biomarkers was an exploratory objective for both cohorts. Bemcentinib inhibited pAXL as well as downstream targets of pAXL (pAKT, pERK and pS6) in longitudinal peripheral blood from patients from the DE cohort (phase 1b part). Intracellular phospho-signaling changes compared to pre-treatment samples indicated changes within 4 h of drug exposure. Inhibition of pAXL in peripheral blood was seen at plasma concentrations of 45–347 ng/mL. A PK-pharmacodynamic analysis of the concentration of bemcentinib in relation to the maximal inhibition of pAXL seen for each patient is presented in Fig. 4. Bemcentinib was shown to inhibit pAXL and its downstream targets in a dose concentration manner generating $EC_{50}$ values in a plasma concentration range of 89–162 ng/mL (Supplementary Fig. 1). The equivalent plasma-free concentration range of 18–32 nM is similar to the expected concentration needed to occupy 80–90% of the AXL receptors of the biological target based on the $K_i$ value for bemcentinib (6 nM).

## Discussion

The treatment of AML patients ineligible for induction chemotherapy remains a challenge. Despite an improvement in response rate with the addition of venetoclax to HMAs or LDAC in 1L (ORR 60–75%), the majority of these patients will relapse and after 1L failure following venetoclax plus HMA, the prognosis of this patient population is dismal with a mOS of 2.9 months[14]. We investigated safety, PK and efficacy of bemcentinib, a potent, selective AXL inhibitor, in a phase 1b trial and found overall good tolerability in an elderly patient population. QTcF was the most relevant treatment-related AE, of which the majority was CTC Grade 1 or 2 which was well manageable with regular ECG monitoring and dose reductions and/or treatment interruptions if necessary. Furthermore, with an ORR of 14% we observed a signal for single-agent efficacy of bemcentinib in R/R elderly AML patients.

Based on these findings, we conducted the first clinical phase 2a study investigating bemcentinib in combination with LDAC in AML patients. The regimen had a comparable safety profile to bemcentinib monotherapy. Common adverse events were primarily those typically associated with AML and treatment with cytarabine (infections, neutropenias, thrombocytopenias) and gastrointestinal (diarrhea, nausea, vomiting). As with the monotherapy, QTc prolongation was observed but was well manageable. Fatal events were mainly caused by infections and were not related to the study drugs. Mutation or inhibition of the tyrosine kinase MERTK, which is closely related to AXL, is associated with serious retinal toxicity[29–31]. However, despite the availability of knockout mice lacking AXL[32], no such association has been described for AXL. Although combined AXL/MERTK inhibitors can induce

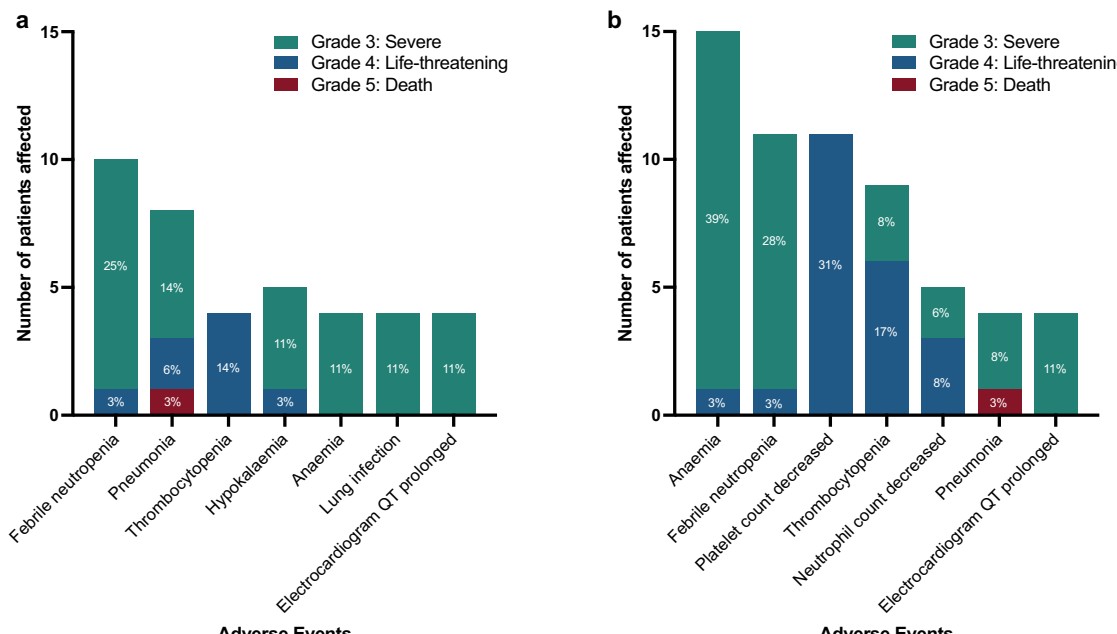

**Fig. 2 | Treatment-emergent adverse events of grade ≥ 3 observed in ≥ 10% of patients. a** Dose escalation cohort. **b** LDAC cohort. Grade 3, severe (green); Grade 4, life-threatening (blue); Grade 5, death (red). Source data are provided as a Source Data file.

retinal toxicity in a mouse model, this toxicity was not observed with a closely related selective AXL inhibitor[33]. Consistent with these previous results, no evidence for retinal toxicities was seen in this trial.

The efficacy results showed an ORR of 25% with a mOS of 8.4 months in the mixed treatment-naïve and R/R AML population. Subgroup analysis demonstrated an ORR of 50% and mOS of 16.1 months in 1 L AML patients ineligible for intensive chemotherapy. Similarly, promising efficacy for the combination of bemcentinib + LDAC was also derived in R/R patients with an ORR of 20% and mOS of 7.8 months. These findings represent a positive signal in light of available 2 L treatments with the caveat that the 1 L treatment landscape in AML has changed during the study conduct. Due to the cohort size of 36 patients, subgroup analyses with respect to efficacy according to type of prior therapy line or specific mutations was not feasible.

Altogether, bemcentinib in combination with LDAC is safe and tolerable in this challenging patient population and evaluation of the therapeutic benefit of this combination in elderly AML patients is warranted in randomized studies. Here, associations with potential predictive biomarkers including genomic alterations could be studied. Furthermore, combination trials with bemcentinib and venetoclax would be of interest in the future.

## Methods

The study complied with all relevant legal and ethical regulations regarding the use of human study participants and was conducted in accordance with the Declaration of Helsinki and the International Council for Harmonisation (ICH) with Good Clinical Practice. Study centers and associated ethics committees/institutional review boards providing approval are listed in Supplementary Table 1. The study was authorized by the Norwegian Medicines Agency (3 July 2014), the Federal Institute for Drugs and Medical Devices (Germany, 21 October 2014), the U.S. Food and Drug Administration (30 October 2014) and the Italian Medicines Agency (7 February 2018). All enrolled patients signed an informed consent form prior to study participation. Patients did not receive compensation. The trial was pre-registered on EudraCT (2014-000165-46, 13 Jan 2014) and ClinicalTrials.gov (NCT02488408). A copy of the pre-registered protocol is provided in Supplementary

Note 1, along with a table detailing the changes in each version (Supplementary Table 2). Following advice from reviewers, the original planned efficacy-evaluable population was abandoned as flawed in favor of evaluating all patients who received study drug.

### Study design

This phase 1b/2a, open-label, multicenter, international trial enrolled 122 patients between 22 October 2014 and 20 July 2021 with database lock in September 2022. The study consisted of a phase 1b dose escalation, followed by a phase 2a cohort expansion. The planned phase 1b followed a standard 3 + 3 study design for bemcentinib dose escalation where the safety and tolerability of bemcentinib in patients with R/R AML was assessed, followed by an expansion to at least 6-10 patients at the selected dose. Due to the introduction of an enhanced formulation, the study was extended with a second dose escalation cohort following a 3 + 3 study design, also with expansion at the selected dose (Fig. 1).

The primary objective for the phase 1b dose escalation cohort was establishment of a maximum tolerated dose, with secondary objectives to identify the DLT profile, to explore safety and tolerability of bemcentinib and to confirm the pharmacokinetic profile of bemcentinib. The conventional 3 + 3 study design gives a 71% chance of escalation if the true but unknown rate of DLT is 20%, and <50% chance of escalation if the true but unknown rate of DLT is >30%. The phase 2a cohort expansion part consisted of 5 different cohorts: bemcentinib as monotherapy in AML (Part B1) or MDS (Part B4), and bemcentinib in combination with LDAC (Parts B2 and B5) or decitabine (Part B3) in AML patients. Here we report efficacy and safety data of the 2 cohorts (B2 and B5) that received bemcentinib in combination with LDAC: cohort B2 was initiated first, enrolling newly diagnosed and R/R AML patients; subsequently, cohort B5 was added, to expand the enrolment in R/R AML patients ineligible for intensive chemotherapy due to advanced age or existing co-morbidities (Fig. 1). Based on a one-sided, within-group test of proportions comparing an anticipated ORR of >20% against the null hypothesis rate of 5%, with power 80% and type I error 0.2 (suitable as evidence of a trend worthy of future study), a sample size of up to 14 evaluable patients was selected for B2. On the same basis, for B5 an initial sample size of 14

**Table 2 | Overview of treatment-emergent adverse events**

| | Dose escalation cohort | | | | | | | | | | | | LDAC cohort | |
| | Bemcentinib 400/100 mg (N=6) | | Bemcentinib 600/200 mg (N=14) | | Bemcentinib 900/300 mg (N=5) | | Bemcentinib 200/100 mg (N=4) | | Bemcentinib 400/200 mg (N=7) | | Overall (N=36) | | Bemcentinib + LDAC 400/200 mg + 20 mg (N=36) | |
| | n (%) | E | n (%) | E | n (%) | E | n (%) | E | n (%) | E | n (%) | E | n (%) | E |
|---|---|---|---|---|---|---|---|---|---|---|---|---|---|---|
| **Treatment-emergent adverse events [1]** | | | | | | | | | | | | | | |
| TEAEs | 5 (83) | 95 | 14 (100) | 196 | 5 (100) | 65 | 4 (100) | 53 | 7 (100) | 168 | 35 (97) | 577 | 36 (100) | 693 |
| SAEs | 4 (67) | 12 | 11 (79) | 21 | 4 (80) | 8 | 4 (100) | 7 | 7 (100) | 23 | 30 (83) | 70 | 28 (78) | 77 |
| Related AEs | 4 (67) | 34 | 7 (50) | 20 | 5 (100) | 31 | 2 (50) | 5 | 7 (100) | 44 | 25 (69) | 134 | 30 (83) | 243 |
| Related SAEs | 1 (17) | 1 | 2 (14) | 2 | 0 (0) | 0 | 1 (25) | 1 | 4 (57) | 5 | 8 (22) | 9 | 14 (39) | 26 |
| **Patients with any TEAE leading to** | | | | | | | | | | | | | | |
| Discontinuation | 2 (33) | 2 | 1 (7) | 2 | 0 (0) | 0 | 0 (0) | 0 | 1 (14) | 1 | 4 (11) | 5 | 4 (11) | 4 |
| Death | 2 (33) | 2 | 2 (14) | 2 | 0 (0) | 0 | 0 (0) | 0 | 3 (43) | 3 | 7 (19) | 7 | 4 (11) | 4 |
| **TEAEs by CTCAE Grade [3]** | | | | | | | | | | | | | | |
| Grade 1: Mild | 5 (83) | 35 | 13 (93) | 64 | 5 (100) | 21 | 4 (100) | 25 | 7 (100) | 74 | 34 (94) | 219 | 33 (92) | 243 |
| Grade 2: Moderate | 4 (67) | 20 | 12 (86) | 52 | 5 (100) | 16 | 4 (100) | 11 | 7 (100) | 42 | 32 (89) | 141 | 32 (89) | 181 |
| Grade 3: Severe | 5 (83) | 17 | 14 (100) | 55 | 5 (100) | 13 | 4 (100) | 10 | 6 (86) | 23 | 34 (94) | 118 | 33 (92) | 121 |
| Grade 4: Life-Threatening | 4 (67) | 4 | 2 (14) | 2 | 1 (20) | 1 | 3 (75) | 3 | 4 (57) | 10 | 14 (39) | 20 | 22 (61) | 42 |
| Grade 5: Death | 2 (33) | 2 | 2 (14) | 2 | 0 (0) | 0 | 0 (0) | 0 | 3 (43) | 3 | 7 (19) | 7 | 4 (11) | 4 |
| **TEAE by relationship to bemcentinib [2,3]** | | | | | | | | | | | | | | |
| Definitely Related | 0 (0) | 0 | 2 (14) | 7 | 1 (20) | 1 | 2 (50) | 2 | 1 (14) | 3 | 6 (17) | 13 | 5 (14) | 12 |
| Probably Related | 1 (17) | 4 | 2 (14) | 4 | 2 (40) | 17 | 1 (25) | 1 | 3 (43) | 11 | 9 (25) | 37 | 9 (25) | 28 |
| Possibly Related | 4 (67) | 21 | 4 (29) | 8 | 5 (100) | 11 | 2 (50) | 2 | 5 (71) | 18 | 20 (56) | 60 | 23 (64) | 89 |
| Unlikely Related | 5 (83) | 22 | 4 (29) | 4 | 4 (80) | 10 | 0 (0) | 0 | 4 (57) | 24 | 17 (47) | 60 | 23 (64) | 162 |
| Unrelated | 5 (83) | 36 | 14 (100) | 171 | 3 (60) | 21 | 4 (100) | 47 | 6 (86) | 102 | 32 (89) | 377 | 35 (97) | 354 |
| Not Applicable | 0 (0) | 0 | 2 (14) | 2 | 2 (40) | 2 | 0 (0) | 0 | 1 (14) | 1 | 5 (14) | 5 | 4 (11) | 4 |
| **TEAE by relationship to cytarabine [2,3]** | | | | | | | | | | | | | | |
| Definitely Related | | | | | | | | | | | | | 4 (11) | 15 |
| Probably Related | | | | | | | | | | | | | 15 (42) | 46 |
| Possibly Related | | | | | | | | | | | | | 24 (67) | 117 |
| Unlikely Related | | | | | | | | | | | | | 21 (58) | 152 |
| Unrelated | | | | | | | | | | | | | 34 (94) | 282 |
| Not Applicable | | | | | | | | | | | | | 7 (19) | 21 |

*Table presents number and percentage of subjects (n(%)) and number of events (E). Percentages are based on the number of subjects in the safety population in each group. AE adverse event, CTCAE Common Terminology Criteria for Adverse Events, E number of events, N, n number of patients (total, affected), SAE serious adverse event, TEAE treatment-emergent adverse event. Headings and subheadings in bold.*
[1] Only treatment-emergent adverse events are included in summary statistics. TEAE period was between the first dose of bemcentinib and 28 days after the last dose. If a subject had multiple occurrences of an AE, the subject was counted only once.
[2] Related refers to those events that were possibly, probably, or definitely related to bemcentinib.
[3] If a subject experienced the same adverse event at more than one severity, or with more than one relationship to study drug, the most severe event and the strongest relationship was chosen.

**Table 3 | Commonly reported SAEs and TEAEs**

| | DE cohort (N = 36) | | | LDAC cohort (N = 36) | | | Overall (N = 72) | | |
|---|---|---|---|---|---|---|---|---|---|
| | n | (%) | E | n | (%) | E | n | (%) | E |
| **SAEs reported in ≥ 10% of patients in either cohort** | | | | | | | | | |
| Febrile neutropenia | 6 | (17%) | 7 | 10 | (28%) | 14 | 16 | (22%) | 21 |
| Pneumonia | 9 | (25%) | 9 | 5 | (14%) | 6 | 14 | (19%) | 15 |
| Lung infection | 4 | (11%) | 4 | 1 | (3%) | 1 | 5 | (7%) | 5 |
| Pyrexia | 2 | (6%) | 3 | 4 | (11%) | 4 | 6 | (8%) | 7 |
| **TEAEs of any grade reported in ≥ 20% of patients in either cohort** | | | | | | | | | |
| Diarrhea | 17 | (47%) | 29 | 17 | (47%) | 34 | 34 | (47%) | 63 |
| Nausea | 12 | (33%) | 15 | 15 | (42%) | 29 | 27 | (38%) | 44 |
| Electrocardiogram QT prolonged | 9 | (25%) | 20 | 17 | (47%) | 33 | 26 | (36%) | 53 |
| Fatigue | 12 | (33%) | 18 | 10 | (28%) | 15 | 22 | (31%) | 33 |
| Anemia | 5 | (14%) | 7 | 16 | (44%) | 29 | 21 | (29%) | 36 |
| Febrile neutropenia | 10 | (28%) | 11 | 11 | (31%) | 20 | 21 | (29%) | 31 |
| Pyrexia | 12 | (33%) | 15 | 8 | (22%) | 9 | 20 | (28%) | 24 |
| Hypokalaemia | 10 | (28%) | 12 | 8 | (22%) | 9 | 18 | (25%) | 21 |
| Oedema peripheral | 7 | (19%) | 10 | 11 | (31%) | 12 | 18 | (25%) | 22 |
| Pneumonia | 10 | (28%) | 10 | 6 | (17%) | 7 | 16 | (22%) | 17 |
| Vomiting | 8 | (22%) | 10 | 8 | (22%) | 10 | 16 | (22%) | 20 |
| Thrombocytopenia | 5 | (14%) | 8 | 10 | (28%) | 11 | 15 | (21%) | 19 |
| Platelet count decreased | 3 | (8%) | 3 | 11 | (31%) | 22 | 14 | (19%) | 25 |
| Dyspnoea | 5 | (14%) | 6 | 9 | (25%) | 9 | 14 | (19%) | 15 |
| Cough | 9 | (25%) | 10 | 5 | (14%) | 7 | 14 | (19%) | 17 |
| Headache | 9 | (25%) | 10 | 4 | (11%) | 5 | 13 | (18%) | 15 |
| Constipation | 3 | (8%) | 3 | 9 | (25%) | 10 | 12 | (17%) | 13 |
| Mouth hemorrhage | 1 | (3%) | 1 | 8 | (22%) | 9 | 9 | (13%) | 10 |

*E* number of events, *AE* adverse event, *N,n* number of patients (total, affected), *SAE* serious adverse event, *TEAE* treatment-emergent adverse event; Headings and subheadings in bold. Sorted by overall frequency in the combined safety population. Where an SAE is ≥10% or a TEAE is ≥20% in one cohort but not the other, it is shown for both cohorts.

relapsed and 14 refractory AML patients was planned. However, upon regulatory review by the FDA, this was reduced to 20 R/R AML patients with 4 refractory AML patients. For this early-stage study, the efficacy analyses are secondary endpoints, and no account was taken of the multiplicity inherent in the assessment of several, presumably non-independent criteria, for ORR.

Results from cohorts B2 and B5 have been amalgamated and are referred to as the LDAC cohort. The primary objective for the LDAC cohort was to assess the safety and tolerability of bemcentinib in combination with LDAC in this patient population. The secondary objective was to characterize the PK profile of bemcentinib and to explore the efficacy of bemcentinib in combination with LDAC, using Objective Response Rate (ORR), disease control rate (DCR), objective response (OR), stable disease (SD) defined as unchanged disease status for 3 treatment cycles, relapse free survival (RFS), event free survival (EFS), and Overall Survival (OS). Exploratory objectives included identification and evaluation of potential biomarkers and assessment of pharmacodynamic biomarkers in tissue and blood.

## Patients
Patients aged >18 years with histological, molecular, or cytological confirmation of AML, unsuitable for intensive chemotherapy due to advanced age ( > 75 years) or co-morbidities (as determined by the local sites due for example to abnormal liver/kidney/heart function or recent prior malignancy), who could receive treatment with

cytarabine and had an Eastern Cooperative Oncology Group (ECOG) performance status of ≤ 2 were eligible to be enrolled into phase 2a (containing cohort B2 and B5, referred to as LDAC cohort). Only R/R AML patients were enrolled into cohort B5, and the number of patients with refractory AML (defined as no hematological response to last AML treatment) and/or patients who had received ≥2 prior lines of treatment for AML, were restricted to 1/3 of the sample size.

Patients with acute promyelocytic leukemia, significant cardiovascular comorbidities, inadequate renal or hepatic function, and candidates for intensive chemotherapy or allogeneic stem cell transplantation were excluded. Prior exposure to gilteritinib was also excluded as it is considered a dual FLT3/AXL inhibitor and may therefore act on the same target as bemcentinib, the drug analyzed in this study. (Full eligibility criteria are available in Supplementary Table 3).

### Study treatment
Bemcentinib was administered orally once daily during continuous 21-day treatment cycles, without rest periods. Treatment was continued for as long as clinical benefit was derived. During the first dose escalation phase (formulation 1), bemcentinib was administered at 400/100 mg, 600/200 mg and 900/300 mg (loading dose administered on Day 1 and Day 2 followed by a daily maintenance dose) in a fed state as monotherapy. In the second dose escalation phase (formulation 2), bemcentinib was administered at 200/100 mg and 400/200 mg (loading dose first 3 days, followed by daily maintenance dose) in fasted patients.

LDAC was administered approximately 30 min after administration of bemcentinib.

Cytarabine was administered subcutaneously at a dose of 20 mg twice daily for 10 days followed by a rest period of at least 18 days depending on persisting myelosuppression.

### Tolerability and safety assessments
Safety assessments including standard clinical laboratory safety tests (hematology, biochemistry, coagulation, and urinalysis), physical examinations, vital signs (blood pressure, heart rate, respiration rate and temperature), and ECOG performance status were performed at baseline and throughout the study. Electrocardiograms were obtained at screening, Cycle 1 Day 1–4, 8, and 15, Cycle 2 Day 1, 8 and 15, at Day 1 of each cycle thereafter, (up to and including Cycle 15), and at Final Study Visit. QT prolongation was assessed using the Fridericia correction $QTcF = QT / \sqrt[3]{RR}$ [34]. Severity of QTcF prolongation was defined as follows: Grade 1: maximum prolongation 450–480 ms; Grade 2: maximum prolongation 481-500 ms; Grade 3: ≥501 ms on at least two separate ECGs; Grade 4: ≥501 ms or >60 ms change from baseline, with Torsades de pointes, polymorphic ventricular tachycardia or signs/symptoms of serious arrythmia; Grade 5: Death.

Adverse events (AEs) were monitored starting at study enrolment until 28 days after end of treatment and graded according to the National Cancer Institute Common Terminology Criteria for Adverse Events (NCI-CTCAE) version 4.0. If drug-related toxicities continued beyond the follow-up period, patients were followed until all drug-related toxicities resolved to grade ≤1, stabilized or returned to baseline. During the dose escalation phase, dose-limiting toxicities were assessed during the first 3 weeks of treatment with bemcentinib (Cycle 1) according to NCI CTCAE, considered unrelated to leukemia progression or intercurrent illness, and defined as any of the following: CTCAE Grade 3 or 4 nausea, vomiting, or diarrhea that persisted despite maximum prophylactic and supportive care; Any other CTCAE Grade 3 or 4 non-hematological toxicity that was considered to be clinically significant and causally related to bemcentinib, excluding isolated changes in laboratory results if no clinical significance or no clinical sequelae and adequately improve within 7 days; Prolonged

**Table 4 | Overview of responses in dose escalation cohort: all patients and subgroups**

| Bemcentinib dose | 400/100 mg (N = 6) n (%) | 600/200 mg (N = 14) n (%) | 900/300 mg (N = 5) n (%) | 200/100 mg (N = 4) n (%) | 400/200 mg (N = 7) n (%) | Overall (N = 36) n (%) |
|---|---|---|---|---|---|---|
| **Non-Responder** | **5 (83%)** | **12 (86%)** | **5 (100%)** | **4 (100%)** | **5 (71%)** | **31 (86%)** |
| NE | 0 | 4 (29%) | 2 (40%) | 1 (25%) | 1 (14%) | 8 (22%) |
| PD | 0 | 3 (21%) | 0 | 1 (25%) | 1 (14%) | 5 (14%) |
| SD | 2 (33%) | 1 (7%) | 1 (20%) | 1 (25%) | 0 | 5 (14%) |
| UC | 3 (50%) | 4 (29%) | 2 (40%) | 1 (25%) | 3 (43%) | 13 (36%) |
| **Responder** | **1 (17%)** | **2 (14%)** | **0 (0%)** | **0 (0%)** | **2 (29%)** | **5 (14%)** |
| CR | 0 | 0 | 0 | 0 | 1 (14%) | 1 (3%) |
| CRi | 0 | 1 (7%) | 0 | 0 | 1 (14%) | 2 (6%) |
| MR | 1 (17%) | 0 | 0 | 0 | 0 | 1 (3%) |
| MLFS | 0 | 1 (7%) | 0 | 0 | 0 | 1 (3%) |
| PR | 0 | 0 | 0 | 0 | 0 | 0 |
| **mOS** | | | | | | |
| mOS, months (95% CI) | 19 (2.6-NA) | 2.5 (1.1-NA) | 6.9 (NA-NA) | 5.9 (4.4-NA) | 8.6 (0.8-NA) | 5.3 (2.6-9.1) |
| **mEFS** | | | | | | |
| mEFS, months (95% CI) | 5.3 (2.6-NA) | 1.2 (0.7-2.1) | NA (NA-NA) | 4.4 (0.5-NA) | 1.6 (0.7-NA) | 2.1 (1.4-2.8) |

CI confidence interval, CR Complete remission, CRi Complete remission with incomplete recovery, mEFS median event free survival, MLFS Morphologic leukemia-free state, mOS median overall survival, MR Marrow response, NA confidence limit could not be determined, NE not evaluable, PD progressive disease, PR Partial response, SD stable disease, UC unchanged. Bold: Headings, subheadings and totals for subcategories below.

**Table 5 | Overview of responses in LDAC cohort: all patients and subgroups**

| Bemcentinib dose 400/200 mg | Treatment-naïve (N = 6) n (%) | Relapsed/ Refractory (N = 30) n (%) | Overall (N = 36) n (%) |
|---|---|---|---|
| **Non-Responder** | **3 (50%)** | **24 (80%)** | **27 (75%)** |
| NE | 0 | 3 (10%) | 3 (8%) |
| PD | 0 | 8 (27%) | 8 (22%) |
| SD | 1 (17%) | 4 (13%) | 5 (14%) |
| UC | 2 (33%) | 9 (30%) | 11 (31%) |
| **Responder** | **3 (50%)** | **6 (20%)** | **9 (25%)** |
| CR | 1 (17%) | 3 (10%) | 4 (11%) |
| CRi | 1 (17%) | 3 (10%) | 4 (11%) |
| PR | 1 (17%) | 0 | 1 (3%) |
| **mOS** | | | |
| mOS, months (95% CI) | 16.1 (1.8-NA) | 7.8 (3.9-11.2) | 8.4 (4.0-15.7) |
| **mEFS** | | | |
| mEFS, months (95% CI) | 16.5 (1.8-NA) | 2.6 (1.3-4.4) | 3.7 (1.8-6.5) |
| **mRFS** | | | |
| mRFS, months (95% CI) | 17.3 (13.7-NA) | 12.6 (2.3-NA) | 13.7 (2.3-28.2) |

CI confidence interval, CR Complete remission, CRi Complete remission with incomplete recovery, mEFS median event free survival, mOS median overall survival, mRFS median relapse free survival, NA confidence limit could not be determined, NE not evaluable, PD progressive disease, PR Partial response, SD stable disease, UC unchanged. Bold: Headings, subheadings and totals for subcategories below.

neutropenia with ANC < 500 and platelet count <75000 after Day 42 from the start of treatment in the absence of residual leukemia; Treatment discontinuation, inability to administer one or more bemcentinib loading dose, or inability to administer three bemcentinib maintenance doses as a result of bemcentinib-related toxicity; Any ventricular arrhythmia.

## Pharmacokinetic assessments

Blood samples for the determination of bemcentinib PK were collected pre-dose and 2-, 4- and 6-hours post-dose Day 1; pre-dose and 6 h post-dose Day 2; pre-dose, 2, 4, 6 and 8 h post dose Day 3; and pre-dose Day 4 in Cycle 1 and weekly in Cycle 2. Thereafter, a pre-dose sample was collected at all study visits ( > Cycle 3 up to and including Cycle 15) and at End of Study. PK samples were measured using a fully validated liquid chromatography with tandem mass spectrometry method for bemcentinib.

A population PK (popPK) model for bemcentinib was derived from data generated across healthy volunteers, patients with solid tumors and patients with myeloid malignancies (283 individuals). The PopPK model was developed by evaluating various structural models including 1 compartment and 2 compartment models. Covariate analysis was performed using a forward addition and backward elimination procedure, from which a preliminary base model was selected. During covariate analysis, continuous covariates were centered on the median population values, and the most common or most relevant category was used as the reference value for categorical covariates. A visual predictive check (VPC) was conducted to verify that the final model adequately predicted both the central tendency and variability of the observed data across studies or by proton pump inhibitor use. Quantification of parameter uncertainty in the final model was assessed by nonparametric bootstrap analysis.

Population PK analysis was performed using the nonlinear mixed effects modelling methodology as implemented in the NONMEM program Version 7.4.4[35]. Data postprocessing was performed using R (Version 3.6.1)[36]. Individual population PK model parameters were estimated using NONMEM with the following $ESTIMATION parameters: METHOD = COND INTERACTION LAPLACE MAXEVAL = 0 POSTHOC NOABORT SIGDIGITS = 3. These parameters were subsequently used to predict individual steady-state profiles with a dense PK sampling scheme using the SIMULATION method in NONMEM. Finally, the predicted PK-time profiles were subjected to a non-compartmental analysis to derive the following exposure parameters: area under the concentration-time curve to the end of the dosing period (AUC$_{0-\tau}$), average plasma concentration and maximum plasma concentration at steady-state (C$_{av,ss}$, C$_{max,ss}$), and half-life (t$_{1/2}$).

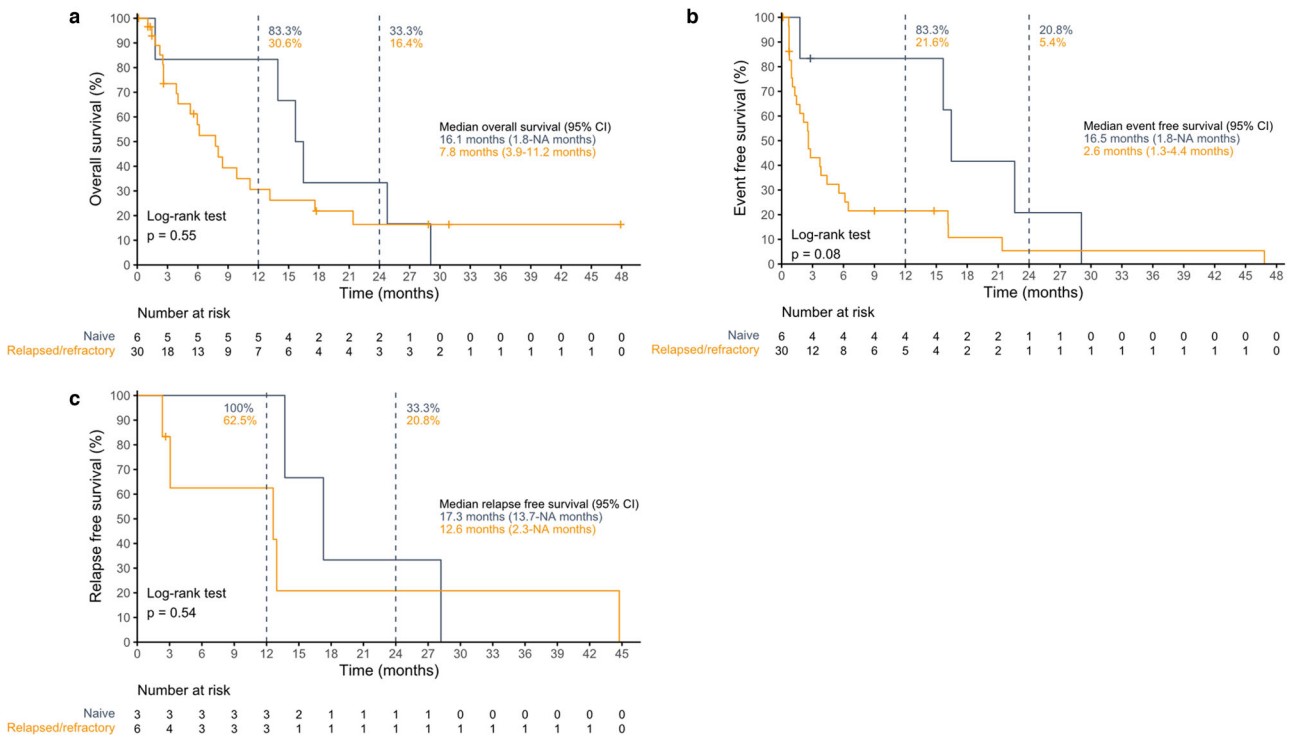

**Fig. 3 | Kaplan-Meier survival curves. a** OS, **b** EFS and **c** RFS for treatment-naive and R/R patients. 95% CI, 95% confidence interval of the median; NA, confidence limit could not be determined. *p*-values from one-tailed log-rank test. Source data are provided as a Source Data file.

## Efficacy assessments

All patients underwent baseline disease assessments at screening (peripheral blood and bone marrow (BM)). Efficacy was assessed according to the blast percentage in a peripheral blood and BM aspirate sample, measurement of absolute neutrophil count, measurement of platelet count, and red cell transfusion requirement according to the European LeukemiaNet[37] response criteria for AML and the modified IWG response criteria for MDS[38]. Response criteria used are summarized in Supplementary Table 4.

Response assessments (peripheral blood and BM) were performed at Cycle 2 Day 1 (pre-dose), Cycle 4 Day 1 (pre-dose) and repeated every 3 cycles thereafter or when clinically indicated. Patients who completed 12 months of study treatment and had not experienced progressive disease (PD) underwent response assessment every 5 cycles until PD was confirmed. Efficacy endpoints were: Objective response rate (ORR), defined as the number of patients with CR, CRi, MLFS, MR or PR divided by the total number of patients in the cohort/group; Disease control rate (DCR), defined as the number of patients with CR, CRi, MLFS, MR, PR or unchanged disease for at least three cycles (considered to have stable disease (SD)) divided by the total number of patients in the cohort/group; Overall survival (OS), defined as (date of death or censoring) – (date of the first study treatment) +1. Censoring date was the last date the patient was known to be alive; Event free survival (EFS), defined as (earliest of date of death, date of progression or date of censoring) – (date of the first study treatment) +1, with event defined as death or progression. Patients who started a new anti-cancer treatment before documented progression were censored at the date of the last assessment prior to start of non-protocol treatment. Patients who were alive without documented progression on the last day of contact were censored at the date of the last assessment; Relapse free survival (RFS), defined as (earliest of date of death, date of relapse or date of censoring) – (date of first assessment where subject had an objective response) +1, only for subjects who achieved an objective response (CR, CRi, MLFS, MR or PR). Patients who started a new anti-cancer treatment before documented

progression were censored at the date of the last assessment prior to start of non-protocol treatment. Patients who were alive without documented progression on the last day of contact were censored at the date of the last assessment.

## Pharmacodynamics

The effects of bemcentinib on pharmacodynamic endpoints of AXL inhibition were determined in BM aspirates and blood samples collected at specified times during the phase 1b (dose escalation) study where bemcentinib was applied as monotherapy.

Pharmacodynamic endpoints included the measurement of inhibition of pAXL, pERK, pAKT, pS6 and pSTAT1/3/5 in blast cells (defined as cells with abnormal surface marker expression as compared to healthy donor peripheral blood mononuclear cells and undefined/immature myeloid cells (defined as cells carrying myeloid markers but lacking distinguishing terminal differentiation markers). Mass cytometry time of flight (CyTOF) was used for the determination of the pharmacodynamic markers using a previously validated and tested panel of mass-tagged antibodies[39,40]. The antibodies in the panel are listed in Supplementary Table 5. Briefly, fixed leukocytes from peripheral blood were barcoded using a 20-plex metal barcoding kit (Standard Biotools, Cell-ID 201060) according to manufacturer's protocol, allowing all the samples from a single individual to be stained and analysed in parallel[41]. All analysis runs included barcoded peripheral blood mononuclear cells (PBMCs) from at least one healthy donor and an identical barcoded reference sample (mix of PBMCs from six healthy donors). Barcoded cells (approx. $6 \times 10^6$) were treated with FC receptor block (Octagam, 1:1000 (Octapharma) in MaxPar cell staining buffer (CSB, Standard Biotools 201068), 15 min, room temperature) and treated with heparin (100 IU/mL in CSB, 20 min) to reduce non-specific staining artefacts[42]. The sample was then stained with an optimised mixture of mass-tagged surface antibodies in a total staining volume of 600 µL (30 min, room temperature). Subsequently, the sample was permeabilized with cold (-20 °C) 100 % methanol (15 min, on ice), re-treated with heparin, and stained with a panel of

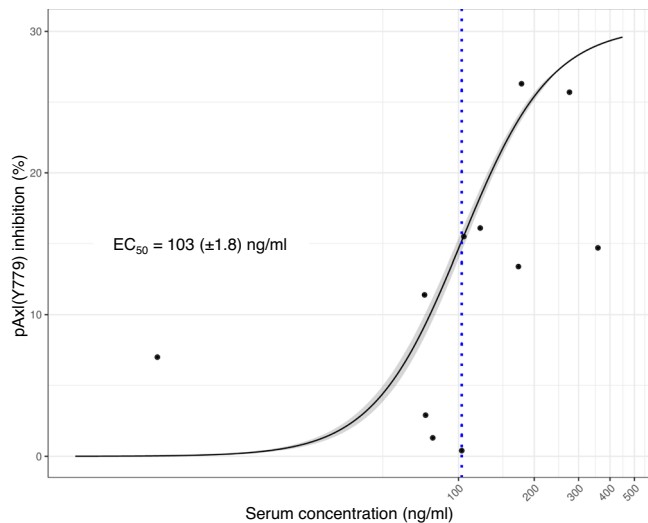

**Fig. 4 | Dose response curve of maximal pAXL inhibition in AML blasts.** AML blasts were identified within the peripheral blood mononuclear cell population by CyTOF and the maximum inhibition observed for each patient was plotted against the bemcentinib serum concentration at the time of maximum inhibition. A robust log-logistic dose response model with lower limit set to 0 was fitted. The calculated $EC_{50}$ (+/- standard error) is indicated (text and dotted line). The black line indicates the fitted model, with the 95% confidence interval indicated by grey shading. $n = 13$ individual patients. Patients for whom maximum inhibition occurred before treatment onset are not shown because the corresponding serum concentration (0 ng/ml) cannot be accommodated on the logarithmic concentration scale. However, they were included for calculation of the dose response curve. Source data are provided as a Source Data file.

intracellular mass-tagged antibodies (as described above). DNA was labelled with iridium-191/193 by incubation in 0.1 nM Ir-nucleic acid intercalator (Standard Biotools 201192) diluted in MaxPar PBS (Standard biotools 201058) containing 4% PFA overnight at 4 °C. Immediately before sample acquisition on a Helios mass cytometer, cells were washed in MaxPar cell staining buffer and MaxPar water (Standard Biotools 201069) before re-suspending in MaxPar water supplemented with a 1:8 dilution of the EQ Four Element calibration beads (Standard Biotools 201078). The acquisition rate was kept below 400 cells per second to limit the number of acquired cell doublets. Machine drift in the data was normalized using the bead normalizer algorithm (Standard Biotools CyTOF Software v7.1). Cell debris and doublets were manually removed by gating on event length and DNA (Ir-191/193). Samples were de-barcoded using the barcode de-convolution tool (Standard Biotools CyTOF Software). Around 13 million cells were clustered using Flowsom (v2.8.0)[43] and ConsensusClusterPlus (v1.64.0)[44]. Twenty-five clusters were merged into 14 clusters identified as B cells, CD4 T cells (Naïve, Memory), CD8 T cells (Naïve, Memory), Monocyte (Classical, Activated Classical, Non-classical), Basophils, Dendritic cells, Granulocytes, NK cells, Undefined/Immature Myeloid cells (lacking distinguishing terminal differentiation markers) and Blasts (defined as cells with abnormal surface marker expression compared to healthy donor PBMCs).

The following steps were used to assess the relationship between marker expression and PK values in the Blasts and Undefined/Immature Myeloid cell populations:

In order to mitigate the diluting effects of signaling in cells unable to respond to bemcentinib, each population of cells was filtered to include only cells expressing AXL (cut-off value = 1). For each population, marker, and sample, the 95th percentile of the non-transformed expression values was calculated. For each patient, the maximum percent inhibition observed was calculated, and a dose response

model (log-logistic with lower limit at 0) was fitted (drc package v3.0-1[45]) for each marker, based on the plasma bemcentinib concentration measured at the time of maximum inhibition. To account for outliers, the lms (robust least median of squares) and the lts (robust least trimmed squares) were used in the calculation of the $EC_{50}$ values where necessary. The modeled $EC_{50}$ ± estimated standard error were reported.

## Statistical analysis

Safety, PK and Efficacy analyses were performed on all patients who had received at least 1 dose of bemcentinib.

Analyses were performed on response data, AEs, and PK parameters. Subgroup analysis by sex was not performed due to the small number of patients. Statistical analyses were performed using descriptive statistics. The time-to-event variables RFS, EFS, and OS were analyzed using Kaplan-Meier estimates, and median time-to-event was reported alongside its 95% confidence interval (CI). Where relevant, differences between different subgroups for time-to-event endpoints were assessed using a one-tailed log-rank test.

## Usage of large language models

We used ChatGPT to format references in the correct style.

## Reporting summary

Further information on research design is available in the Nature Portfolio Reporting Summary linked to this article.

## Data availability

Source Data are provided with this paper. Access to patient-level data is restricted due to data privacy laws but may be requested by completing a data access agreement, available on application to the data access committee (Data-access@bergenbio.com). Access will be granted exclusively to qualified investigators for appropriate non-commercial use that is expected to lead to a publication. Access will be subject to approval by a regional ethical committee to ensure that it is in line with lawful basis for processing, data protection regulations, and ethical standards. We aim to provide an initial response to data access requests within 3 weeks. Correspondence and materials requests should be addressed to S.L. (Sonja.Loges@medma.uni-heidelberg.de) and C.O. (cristina.oliva@bergenbio.com). All remaining data can be found in the Article, Supplementary and Source Data files. Source data are provided with this paper.

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

## Acknowledgements

We thank all the patients, investigators, and study coordinators who participated in and supported this study. BerGenBio provided financial support for the study and participated in the design, study conduct, analysis, and interpretation of data, as well as the writing, review, and approval of the manuscript. Abbas Shivji and Oliver Dewhirst, who are employees of Exploristics UK, provided help with the generation of fig-ures and tables. Funding was received from the following grants: Grant Agreement No. 758713, European Union's Horizon 2020 research and innovation program, S.L.; Hector Stiftung II (no grant number available), S.L.; Project 445549683, GRK 2727/1 InCheck, Deutsche For-schungsgemeinschaft, S.L., I.B-B.

## Author contributions

S.L. and B.T.G. conceived and designed the study. S.L., M.H., J.C., G.S., S.K.-S., M.Crugnola, N.D.R., R.L., D. Mattei, W.F., Y.A.V. and B.T.G. contributed study materials and patients. S.L., M.H., J.C., G.S., S.K.-S., M.Crugnola, N.D.R., R.L., D. Mattei, W.F., Y.A.V., D. Micklem, L.H.N., N. Madeleine, N. McCracken, C.O., C.G.-C. and B.T.G. collected and assembled study data. S.L., I.B.-B., L.-M.R., M.J., C.D.I., N.B., J.W., M. Collienne, D. Micklem, L.H.N., N. Madeleine, N. McCracken, C.O., C.G.-C and B.T.G. analysed and interpreted data. N.B. provided administrative support. All authors co-wrote and approved the manuscript. All authors had access to and reviewed the clinical trial results.

## Funding

## Competing interests

S.L. received travel support, advisory board honoraria and research funding from BerGenBio. She is currently supported by the European Research Council (ERC) under the European Union's Horizon 2020 research and innovation program (Grant Agreement No. 758713) and by the Hector Stiftung II. M.H. is a consultant, advisor and/or speaker at AbbVie, Amgen, Bristol Myers Squibb, Certara, Glycostem, Jazz, Janssen, LabDelbert, Novartis, Pfizer, PinotBio, Servier and Sobi. His institution received research support from AbbVie, Agios, Astellas, BerGenBio, Bristol Myers Squibb, Glycostem, Jazz, Karyopharm, Loxo Oncology and PinotBio. D. Micklem, L.H.N., N. Madeleine, N. McCracken, C.O., C.G.-C. are employees at BerGenBio and may hold BerGenBio stock or stock options. The other authors declare no competing financial interests.

## Additional information

Sonja Loges [1,2,3] ✉, Michael Heuser [4,5], Jörg Chromik[6], Grerk Sutamtewagul [7], Silke Kapp-Schwoerer[8], Monica Crugnola[9], Nicola Di Renzo[10], Roberto Lemoli[11,12], Daniele Mattei[13], Walter Fiedler[14], Yesid Alvarado-Valero[15], Isabel Ben-Batalla[1,2,3], Jonas Waizenegger[1,2,3], Lisa-Marie Rieckmann[1,2,3], Melanie Janning[1,2,3], Maike Collienne [1,2,3], Charles D. Imbusch [16,21], Niklas Beumer [1,2,3,16,17,21], David Micklem [18], Linn H Nilsson[18], Noëlly Madeleine[18], Nigel McCracken[19], Cristina Oliva[19], Claudia Gorcea-Carson[19] & Bjørn T. Gjertsen[20]

[1]German-Cancer-Research-Center-(DKFZ)-Hector Cancer Institute, University Medical Center Mannheim, Mannheim, Germany. [2]Division of Personalized Medical Oncology (A420), German Cancer Research Center (DKFZ), Heidelberg, Germany. [3]Department of Personalized Oncology, University Hospital Mannheim, and Medical Faculty Mannheim, University of Heidelberg, Mannheim, Germany. [4]Hematology, Hemostasis, Oncology and Stem Cell Transplantation, Hannover Medical School, Hannover, Germany. [5]Comprehensive Cancer Center Niedersachsen, Hannover Medical School, Hannover, Germany. [6]University Hospital Frankfurt, Frankfurt, Germany. [7]University of Iowa Hospitals and Clinics, Iowa City, IA, US. [8]University Hospital of Ulm, Ulm, Germany. [9]University of Parma, Parma, Italy. [10]Haematology and Stem Cell Transplantation Unit, Vito Fazzi Hospital, Lecce, Italy. [11]Department of Internal medicine (DIMI), University of Genoa, Genoa, Italy. [12]IRCCS-San Martino Hospital, Genoa, Italy. [13]Azienda Sanitaria Ospedaliera (ASO) Santa Croce e Carle, Cuneo, Italy. [14]University Medical Center Hamburg-Eppendorf, Hamburg, Germany. [15]The University of Texas M.D. Anderson Cancer Center, Houston, TX, US. [16]Division of Applied Bioinformatics (B330), German Cancer Research Center (DKFZ), Heidelberg, Germany. [17]Faculty of Biosciences, Heidelberg University, Heidelberg, Germany. [18]BerGenBio ASA, Bergen, Norway. [19]BerGenBio Ltd, Oxford, UK. [20]Haukeland University Hospital, Bergen, Norway, & Centre for Cancer Biomarkers (CCBIO), Department of Clinical Science, University of Bergen, Bergen, Norway. [21]Present address: Institute of Immunology, University Medical Center Mainz, Mainz, Germany and Research Center for Immunotherapy (FZI), Mainz, Germany. ✉e-mail: s.loges@dkfz-heidelberg.de

