## [Transparent Peer Review file · Nature Communications]

Bemcentinib as Monotherapy and in Combination with Low-Dose Cytarabine in Acute Myeloid Leukemia Patients Unfit for Intensive Chemotherapy: a phase 1b/2a trial

Corresponding Author: Professor Sonja Loges

Version 1:

Reviewer comments:

Reviewer #1

(Remarks to the Author)

This is a well written and comprehensive manuscript by Loges et al reporting on the safety and activity of an oral AXL inhibitor, Bemcentinib, in a dose escalation phase 1 study in R/R AML and an expansion cohort in combination with LDAC. As the authors point out, there is a high unmet need for novel therapies and combinations in R/R AML, particularly in patients who relapse after HMA/Ven frontline therapy. Inhibition of AXL is a rational and plausible mechanism of action in AML and thus there is merit to this study.

I have several comments and suggestions that would strengthen this manuscript.

1) Abstract- I would recommend defining LDAC.

- The sentence starting with "In the 32 efficacy-evaluable..." should be broken up into 2 separate statements.
- It is unclear what survival benefit is alluding to in abstract since this is not a randomized study or being statistically compared to any historical controls. I would therefore either remove this statement or revise.

2) Introduction

- Based on Maiti et al, Haematologica 2021, median OS of patients with R/R AML after HMA/Ven is 2.4 months (not 2.9 months).

3) Methods

- Cohort B2 vs. B5 is confusing. I would recommend these cohorts be combined for the actual data and results as it is confusing to list them as separate cohorts throughout the paper.
- Why was LDAC dose different in Cohort B2 and B5?

4) Results

- I would recommend combining Tables 1 and 2 with the same patient characteristics.
- Did any patient receive prior Venetoclax in either cohort?
- I would also include most common mutations seen pre-treatment in Table
- Safety/Tolerability
- I would not recommend starting this section with Bemcentinib was well tolerated. That should be in the Discussion.
- How many patients required dose reductions or discontinuations due to AE's in monotherapy?
- Please specify whether QT Prolongation specifically refers to QTcFridericia and the specific grading for all AE's listed. At what time point on study did patients experience these AE's?
- Last paragraph on page 7 should be omitted and should just reference Supp. Table 1.
- Last paragraph on page 8 should also be omitted starting with "Overall, the mean vital signs..."
- QTcF events paragraph on page 9 is confusing as it is worded. It should be reported overall Grade of QTcF prolongation, at what time point on the study was this seen, and were these related or unrelated. Echocardiogram statement is confusing- did all patients get follow up echocardiograms? QTcF prolongation would not imply other cardiac toxicities seen and if this was not uniformly done then this statement is misleading.
- I would also not state in the Results that Bemcentinib in combination with LDAC was generally well tolerated (this should

be in Discussion).

- What were the AEs that led to dose interruptions or reductions in combination?

- Treatment Efficacy

- As treatment naive (only n=6) and R/R are very different pt populations, these should be separated from each other when assessing activity. For instance, it is not informative to list overall ORR and CR from the entire combination cohort. I would only recommend showing the breakdown of Tx-naive and R/R AML patients. This should impact the responders and non-responders assessment as well.

- DCR is not a well validated or commonly used treatment assessment in AML and not sure how informative this is.

- Are there subgroups of patients who are enriched in responders or had longer duration of response? What is missing is genomic data on response evaluable patients. Although this is a small study not powered to show effect sizes, hypothesis generating information can be gathered from this data.

5) Figures

- Figures 3 and 4 should be combined. Some subfigures can be omitted- i.e, Figure 3A, B, E and F- as it is most relevant to differentiate Tx-naive from R/R as mentioned above.

6) Discussion

- It is confusing to state that patients receiving 2L have a survival advantage since this is not a statistically compared treatment group and overall outcomes of AML patients are cumulatively worse with later lines of therapy.

- Would remove survival benefit section in responders since this is confusing and could be misleading. It is okay to state that responders had numerically higher survival rates (as would be expected).

- Would remove "very favorable"

- Manuscript would be strengthened adding data in Discussion of patient population that may have benefitted the most beyond just R/R 2L.

- Future directions? What do the authors propose is the next step?

Reviewer #2

(Remarks to the Author)

General comments:

1. This is a clinical trial, please justify your sample size.

1.1. In dose escalation cohort, 5 dose levels were considered but how are they being escalated (i.e. in what order)?

1.2. Supplement Figure 2 appears to be a multiple dose level simultaneously and not in a sequence.

1.3. If this is a 3+3 design, the 600/200 cohort does not have any patient experienced DLT so why the cohort was fully enrolled to 6 patients. Similarly, why did 600/200 cohort enrolled 14 patients which is not usual for a 3+3 design.

1.4. Was the sample size calculation done for any of the cohort B2 and B5? If not, please justify why not.

2. Method section

2.1. what were the dose limiting toxicity? It should be clearly described.

2.2. Line 453, does NONMEM stand for "non-linear mixed-effect modeling"? If so, please introduce the abbreviation.

2.3. Line 456, "MAXEVAL=0" is not a method. It is an option that you set. Please explain that option or provide actual method.

2.4. Line 460, please define all the exposure parameters.

2.5. Line 473, please define all the efficacy endpoints.

2.6. Line 493, please justify the choose of limiting efficacy analysis population to those with at least 1 post-baseline assessment. This restriction will bias the results.

2.7. Line 497, please explain why ORR and DCR are classified as time-to-event endpoints.

3. The reporting can be confusing

3.1. I thought this paper focuses on cohort B2 and B5 only (line 370) for efficacy but line 239 described a cohort with bemcentinib as monotherapy (which turns out to be the phase 1a patients). I don't think it is fair to lump the phase 1 patients together for reporting efficacy since they received very different dose. Please justify.

Specific comments:

1. Line 65, LDAC abbreviation was used but not introduced.

2. Line 93-95, I have trouble understanding the sentence. The median OS is 4 months after R/R on first line but I don't know what that 2.9 months referred to.

3. In cohort B2, how many patients were newly diagnosed and how many were R/R AML? Can you add that information in Figure 1 or Figure 1 legend?

4. Line 381. I fail to see how you can evaluate potential predictive biomarkers when all patients were treated with the same agents. Please explain.

5. Line 277-278, was the half life reported correctly? i.e. the lowest dose has the longest half-life?

Reviewer #3

(Remarks to the Author)

In this manuscript, the authors present data from phase 1/2 study on AML/HR MDS patients treated by an Axl inhibitor, bemcentinib, as single agent or in association with LDAC.

R/R and unfit AML are bad prognosis population with a high unmet medical need.

AXL is a tyrosine kinase receptor with adhesion and environmental sensing capacities. Through its canonical ligand GAS6, it is activated by bypassing peripheral apoptotic bodies and the downstream pathways give it a role in cell survival and proliferation, as well as in migration and efferocytosis. Relatively ubiquitously expressed, Axl belongs to RTAMs, that generally play a major role in the immune system, particularly in the innate immune system. Axl plays also a major role in cancer resistance to treatment, in AML as in various others cancer, and many data support its association with other proapoptotic drugs to enhance their efficacy in a synergistic manner.

The topic is of interest, but the manuscript suffers from some shortcomings that prevent its publication in its current form.

MAJOR

Results

- The patient disposition diagram should be provided with screen failure, figure should not be in supp. In contrast, for an early phase clinical trial, there are a lot of OS curves.
- Characteristics for patients should include cytogenetic risk in both Tables 1 and 2, molecular profile for most prevalent/important mutated genes (at least NPM1, FLT3, TP53) and also previous allo-HSCT and previous exposition to venetoclax or intensive chemotherapy for R/R cohort
- For fatal TEAEs, please precise if event occurred in patients with active disease or in CR
- At the beginning of ASP2215/gilteritinib development, there was a warning about Axl inhibition and eyesight problems due to RTAM role that has been demonstrated in MERTK^{-/-} mice with retinitis pigmentosa, since the cells of the retinal epithelium can no longer ingest apoptotic bodies, inducing inflammation and retinal degeneration. Please provide a specific sentence in ophthalmological AE, even if there is not.
- Treatment efficacy: % of patients bridged to transplant should be provided. Indeed, median OS in R/R disease for patient unfit for intensive chemotherapy and with relatively low CR/CRi rate are surprisingly long
- Please provide median time to ANC recovery, platelet recovery, time to CR etc.

MINOR

Title

Minor: Intensive chemotherapy would be more informative for NC readers

Abstract

Minor: line 59, AML and HR MDS would be more precise

Minor: abbreviation w/o definition

Minor: last sentence about overall efficacy in "second line" should be replaced by objective data.

Introduction

Minor: line 88, ref (Burnet Cancer 2007, LDAC in unfit AML) is missing

Minor: several abbreviations, examples OS line 100 previously defined line 93

Methods

Minor: line 386, reasons for unsuitability for intensive chemotherapy should be defined

Minor: line 394, explain why exclusion of prior treatment of gilteritinib, all NC readers doesn't know that it is a dual FLT3/AXL inhibitor

Minor: Study treatment line 410, please provide an administration schedule. Is there a rest between D21 bemcentinib and D28 LDAC? or bemcentinib is given continuously in D21-cycle and LDAC for 10 days in D28-cycles?

Minor: Efficacy assessment: Ref is missing for IWG and author should also discuss why they do not use ELN response criteria for AML that represent 94% of the cohort. CR and CRi are presented in Table and should be detailed here, same thing for ORR, DCR.

Minor: Please provide the way to calculate QTc (Fridericia ?)

Results

Minor: Cohort A does not refer to anything clearly defined at this stage, "dose escalation cohort" should be used

Minor: Cytogenetic risk in Table 2 should be defined (Grimwade, Blood 2010?) and paper should be referenced line 155 or in Methods

Minor: for all tables, abbreviations are missing

Minor: line 164, the introduction "bemcentinib as monotherapy was generally well-tolerated" is a conclusion. The sentence should be deleted.

Minor: line 193, a "." is surplus.

Minor: line 219, the introduction "bemcentinib in combination with LDAC was generally well-tolerated" is a conclusion. The sentence should be deleted.

Version 2:

Reviewer comments:

Reviewer #1

(Remarks to the Author)

Thank you for revising the manuscript based on my comments.

I just have several minor comments that would improve the manuscript.

In the Abstract- I would recommend clarifying that dose escalation was done as a single agent in R/R AML and that there was an expansion cohort of combination therapy with LDAC in both newly diagnosed and R/R AML. It is confusing that

clinical activity is reported in both newly diagnosed and R/R AML in the abstract without further clarification about eligibility and study design.

I would also revise the last statement in the Abstract and not state a "positive signal" was observed and just state that these results are promising and warrant further investigation.

Reviewer #2

(Remarks to the Author)

Most of my previous comments were addressed to my satisfaction. However, there are still a few issues that deserve authors' attention.

1. The sample size assumption is the same for B2 and B5 but the actual sample size are different. I understand that the cohort is exploratory but please provide a rationale why the sample sizes are different.
2. Line 316. What is "a test of size 0.2"? Do you mean a "one-sided type I error of 0.2"?
3. Line 326-329. All endpoints should be clearly defined. This was pointed out previously but was not updated sufficiently.
 - 3.1.1. For ORR and DCR, the patients who are evaluable for the endpoint (i.e. the denominator) and the definition of success (i.e. the numerator) should be detailed.
 - 3.1.2. For RFS, EFS, OS, the event/censor should be defined. For example, what is the "events" considered for event-free survival? What is the start time, registration or first dose?
4. Figure 1, things do not add up. 166 patients were screen and 43 were excluded which implies 123 were enrolled but the manuscript text states 122 were enrolled.
5. Figure 1, the right-hand side box of "phase2a, other cohorts". Should the cohorts listed be B4 instead of B5?

Reviewer #3

(Remarks to the Author)

All my concerns have been addressed in the revisions.

Version 3:

Reviewer comments:

Reviewer #1

(Remarks to the Author)

Thank you for addressing my comments. I have no further comments.

Reviewer #2

(Remarks to the Author)

Thank you for addressing all my comments.

GENERAL REMARK: The authors would like to thank all reviewers for the effort they put into reviewing this paper and for their insightful comments. As requested, we have addressed all concerns and critiques of the referees in a point-by-point response.

RESPONSE TO REVIEWER #1

This is a well written and comprehensive manuscript by Loges et al reporting on the safety and activity of an oral AXL inhibitor, Bemcentinib, in a dose escalation phase 1 study in R/R AML and an expansion cohort in combination with LDAC. As the authors point out, there is a high unmet need for novel therapies and combinations in R/R AML, particularly in patients who relapse after HMA/Ven frontline therapy. Inhibition of AXL is a rational and plausible mechanism of action in AML and thus there is merit to this study.

Authors would like to thank the reviewer for this positive feedback.

I have several comments and suggestions that would strengthen this manuscript.

Abstract

1.1 Abstract- I would recommend defining LDAC.

We have added the definition (see line 62 of the revised manuscript).

1.2 The sentence starting with "In the 32 efficacy-evaluable..." should be broken up into 2 separate statements.

The sentence is now split into two sentences (see lines 68ff of the revised manuscript).

1.3 It is unclear what survival benefit is alluding to in abstract since this is not a randomized study or being statistically compared to any historical controls. I would therefore either remove this statement or revise.

Authors would like to apologize for this overstatement and agree that it is impossible to statistically compare our data with other datasets. We have revised the sentence in the abstract to "The 7.8 month overall survival observed in R/R patients represents a positive signal warranting further validation" (see lines 75ff of the revised manuscript).

Introduction

1.4 - Based on Maiti et al, Haematologica 2021, median OS of patients with R/R AML after HMA/Ven is 2.4 months (not 2.9 months).

We agree with the reviewer that the median OS for all patients who failed VEN+HMA was indeed 2.4 months. However, we believe that it is more appropriate to use the median OS for the subset of patients who received salvage therapy (2.9 months) in the context of our trial. We have added text to the introduction to clarify that we are referring to this subset (see lines 97f of the revised manuscript).

Methods

1.5 - Cohort B2 vs. B5 is confusing. I would recommend these cohorts be combined for the actual data and results as it is confusing to list them as separate cohorts throughout the paper.

Following the reviewer's suggestion, we have combined the data for Cohorts B2 and B5 and refer to the combined cohort throughout as the LDAC Cohort. Similarly, we have removed references to "Cohort A", replacing with "dose escalation cohort" or "DE cohort". We hope this wording is clearer and makes the manuscript easier to read.

1.6 - Why was LDAC dose different in Cohort B2 and B5?

Authors would like to apologize for the confusion. Cytarabine was administered twice daily at 20mg flat dose in both cohorts (B2 & B5). The 20 mg/m² as stated in the original version in line 424 was an error. We corrected this mistake in the revised version of the manuscript (see lines 370ff of the revised manuscript).

Results

1.7 - I would recommend combining Tables 1 and 2 with the same patient characteristics.

Following this recommendation Tables 1 and 2 have been combined (now Table 1), and additional information as requested has been incorporated into the combined table.

1.8 - Did any patient receive prior Venetoclax in either cohort?

No patients from the dose escalation cohort received prior Venetoclax, but 11 patients in the LDAC cohort received prior Venetoclax. We have included this information into Table 1.

1.9 - I would also include most common mutations seen pre-treatment in Table

We have added information on FLT3 status to Table 1. This was the only common mutation for which data was collected.

Safety/Tolerability

Authors would like to inform the reviewer that we have significantly revised the Safety/Tolerability section to improve clarity and we have included additional information requested by the reviewers.

1.10 - I would not recommend starting this section with Bemcentinib was well tolerated. That should be in the Discussion.

This statement has been removed (see line 145 of the revised manuscript).

1.11 - How many patients required dose reductions or discontinuations due to AE's in monotherapy?

We have added text (see lines 185ff of the revised manuscript) and included all AEs leading to dose modifications into Supplementary Table 5 of the revised manuscript.

1.12 - Please specify whether QT Prolongation specifically refers to QTcFridericia and the specific grading for all AE's listed. At what time point on study did patients experience these AE's?

All mentions of QT prolongation refer to QTcFridericia prolongation. We have updated the text accordingly (see, for instance, line 72 of the revised manuscript). We have added also a supplementary table (Supplementary Table 3) detailing the severity and timing of QTcF events.

1.13 - Last paragraph on page 7 should be omitted and should just reference Supp. Table 1.

As suggested, the information from this paragraph has been replaced with a reference to the corresponding supplementary tables. Data on adverse events are now listed in Supplementary Tables 2-5. (see lines 156ff of the revised manuscript).

1.14 - Last paragraph on page 8 should also be omitted starting with "Overall, the mean vital signs..."

The corresponding paragraph has been deleted from the manuscript.

1.15 - QTcF events paragraph on page 9 is confusing as it is worded. It should be reported overall Grade of QTcF prolongation, at what time point on the study was this seen, and were these related or unrelated. Echocardiogram statement is confusing- did all patients get follow up echocardiograms? QTcF prolongation would not imply other cardiac toxicities seen and if this was not uniformly done then this statement is misleading.

The description of QTcF events has been rewritten, with the requested information collected in a new supplementary table (Supplementary Table 3).

Per the protocol, echocardiograms were to be uniformly collected at screening and at C4D1. However, as we did not observe signs of cardiotoxicity here, we removed the statement on echocardiograms from the revised manuscript for consistency reasons because we do not comment on all unchanged clinical parameters related to the study.

1.16 - I would also not state in the Results that Bemcentinib in combination with LDAC was generally well tolerated (this should be in Discussion).

As advised, we have removed this statement from the results section. The revised Safety and Tolerability section (see lines 145ff of the revised manuscript) no longer includes it.

1.17 - What were the AEs that led to dose interruptions or reductions in combination?

We have added text (see lines 185ff of the revised manuscript), and a table (Supplementary Table 5) detailing all TEAEs leading to dose modifications.

Treatment Efficacy

1.18 - As treatment naive (only n=6) and R/R are very different pt populations, these should be separated from each other when assessing activity. For instance, it is not informative to list overall ORR and CR from the entire combination cohort. I would only recommend showing the breakdown of Tx-naive and R/R AML patients. This should impact the responders and non-responders assessment as well.

We thank the reviewer for this insightful comment and have substantially revised the text and the corresponding figure. We now report outcome data only stratified by whether patients were treatment-naive or R/R (see, for instance, Figure 3 of the revised manuscript).

1.19 - DCR is not a well validated or commonly used treatment assessment in AML and not sure how informative this is.

Although we agree with the reviewer that DCR is not commonly used, it was included in the protocol as a secondary endpoint, and we therefore think that it is appropriate to include in this manuscript.

1.20 - Are there subgroups of patients who are enriched in responders or had longer duration of response? What is missing is genomic data on response evaluable patients. Although this is a small study not powered to show effect sizes, hypothesis generating information can be gathered from this data.

Unfortunately, as genomic data besides the *FLT3* status was not collected for these patients per protocol, we are unable to perform this subgroup analysis. While we agree that it could have been interesting, the small study size was in any case not powered to allow such subgroup analysis.

Figures

1.21 - Figures 3 and 4 should be combined. Some subfigures can be omitted- i.e, Figure 3A, B, E and F- as it is most relevant to differentiate Tx-naive from R/R as mentioned above.

The two figures have now been combined to Figure 3 and we have substantially reduced the number of panels shown, focusing only on the analyses stratifying patients by whether they were treatment-naive or R/R.

Discussion

1.22 - It is confusing to state that patients receiving 2L have a survival advantage since this is not a statistically compared treatment group and overall outcomes of AML patients are cumulatively worse with later lines of therapy.

We have substantially revised the discussion due to the suggested removal of comparisons other than naïve vs R/R. We are no longer referring to survival advantage throughout the revised manuscript.

1.23 - Would remove survival benefit section in responders since this is confusing and could be misleading. It is okay to state that responders had numerically higher survival rates (as would be expected).

We have substantially revised the discussion due to the suggested removal of comparisons other than naïve vs R/R. We are no longer referring to survival advantage throughout the revised manuscript.

1.24 - Would remove "very favorable"

We have rephrased the corresponding statement, thereby removing the phrase "very favorable" (see line 275 of the revised manuscript).

1.25 - Manuscript would be strengthened adding data in Discussion of patient population that may have benefitted the most beyond just R/R 2L.

As discussed under point 1.20, we unfortunately do not have the genomic data available that would be relevant for such a discussion, and the number of patients is very small.

1.26 - Future directions? What do the authors propose is the next step?

Following this valuable suggestion, we have included a section on future directions (see lines 280ff of the revised manuscript).

RESPONSE TO REVIEWER #2

General comments:

2.1 - This is a clinical trial, please justify your sample size.

We have added justification of the sample sizes for both cohorts (see lines 302ff and lines 312ff of the revised manuscript).

2.2 - In dose escalation cohort, 5 dose levels were considered but how are they being escalated (i.e. in what order)?

Based on this question, we included information about the dose escalation into the Study Design section and clarified how they were escalated both in the text (see lines 288ff of the revised manuscript) and in the revised Figure 1.

2.3 - Supplement Figure 2 appears to be a multiple dose level simultaneously and not in a sequence.

Authors would like to refer the reviewer also to our reply to comment # 2.2. We have revised the original Supplementary Figure 2 to reflect the sequence and order of dose

escalation. Based on another reviewer's comment, this figure is now moved into the main manuscript as Figure 1.

2.4 - If this is a 3+3 design, the 600/200 cohort does not have any patient experienced DLT so why the cohort was fully enrolled to 6 patients. Similarly, why did 600/200 cohort enrolled 14 patients which is not usual for a 3+3 design.

The trial was originally a classical 3+3 design, with a planned expansion at the selected dose to "a minimum of 6-10 patients". The expansion of the 600/200 dose reflects the original intention to use this as the selected dose. However, based on subsequent PK results from a different trial with bemcentinib, the protocol was modified to test an additional two doses in a modified formulation. This led to selection of 400/200 as the phase 2a dose, and expansion of this group to the "minimum of 6-10 patients" (see lines 288ff of the revised manuscript).

(1.4) 2.5 - Was the sample size calculation done for any of the cohort B2 and B5? If not, please justify why not.

A sample size calculation was done for cohorts B2 and B5, and we have added a description within the Study Design section of the revised manuscript (see lines 312ff of the revised manuscript).

Method section

2.6 - What were the dose limiting toxicity? It should be clearly described.

A description of what adverse events were considered dose-limiting toxicities is provided in Supplementary Methods 3. We have edited the text to clarify this (see line 386 of the revised manuscript).

2.7 - Line 453, does NONMEM stand for "non-linear mixed-effect modeling"? If so, please introduce the abbreviation.

The reviewer is correct that NONMEM frequently stands for "non-linear mixed-effect modeling". However, NONMEM (not as an abbreviation) is also the name of a software package used to perform such modeling, and it is in this sense that it is used here (see line 397 and Supplementary Methods 4 of the revised manuscript).

2.8 - Line 456, "MAXEVAL=0" is not a method. It is an option that you set. Please explain that option or provide actual method.

Authors thank the reviewer for pointing out this issue. The corresponding methods text has now been revised to provide more details on the options used. To reduce the length of the manuscript, details of the derivation of the exposure parameters have been moved to the Supplementary Methods (see Supplementary Methods 4 of the revised manuscript).

2.9 - Line 460, please define all the exposure parameters.

The manuscript text now contains definitions for all the exposure parameters (see lines 398ff of the revised manuscript).

2.10 - Line 473, please define all the efficacy endpoints.

The manuscript text now includes the definitions for these efficacy endpoints (see lines 411ff of the revised manuscript).

2.11- Line 493, please justify the choose of limiting efficacy analysis population to those with at least 1 post-baseline assessment. This restriction will bias the results.

We thank you for the comment on the analysis populations and acknowledge that this selection of efficacy analysis population could have biased the results. We have therefore re-analyzed all efficacy endpoints using the entire Safety Analysis population (all patients who received at least one dose of study drug) and revised the manuscript accordingly. This reanalysis has however only led to minor differences in the results, suggesting that the bias was not too serious. However, we agree that it is more accurate to present data for all patients that received bemcentinib.

2.12 - Line 497, please explain why ORR and DCR are classified as time-to-event endpoints.

ORR and DCR were mistakenly classified as time-to-event endpoints and have been removed from the list of time-to-event variables (see line 433 of the revised manuscript).

Results

The reporting can be confusing

2.13 - I thought this paper focuses on cohort B2 and B5 only (line 370) for efficacy but line 239 described a cohort with bemcentinib as monotherapy (which turns out to be the phase 1a patients). I don't think it is fair to lump the phase 1 patients together for reporting efficacy since they received very different dose. Please justify.

Authors would like to thank the reviewer for pointing this out. The paper originally focused on the LDAC (B2 and B5) cohort exclusively, but the dose-escalation monotherapy cohort was then added following the editor's request to include this information. We regret that the study design section was not sufficiently updated following this change. We have rewritten the study design section (see lines 288ff of the revised manuscript) based on this and other comments and hope that it is now clearer.

We appreciate that the dose-escalation cohort patients received different doses of bemcentinib and have reported response rates per group in the relevant table (Table 4) and also, for the 400/200mg group, in the text (see lines 194ff of the revised manuscript).

Specific comments:

2.14 - (Specific comment 1) Line 65, LDAC abbreviation was used but not introduced.

We have now introduced the abbreviation at the appropriate position prior to where this issue occurred (see line 62 of the revised manuscript).

2.15 - (Specific comment 2) Line 93-95, I have trouble understanding the sentence. The median OS is 4 months after R/R on first line but I don't know what that 2.9 months referred to.

The OS of 4 months refers to all R/R patients while the OS of 2.9 months refers to the sub-group that received venetoclax-based therapy as first-line therapy. We have rephrased this sentence in order to state this more clearly (see lines 96ff of the revised manuscript).

2.16 - (Specific comment 3) In cohort B2, how many patients were newly diagnosed and how many were R/R AML? Can you add that information in Figure 1 or Figure 1 legend?

We have added this information to Figure 1. Furthermore, we would like to inform the reviewer that based on request of reviewer #1 we now report outcome data stratified by whether patients were treatment-naive or R/R.

2.17 - (Specific comment 4) Line 381. I fail to see how you can evaluate potential predictive biomarkers when all patients were treated with the same agents. Please explain.

In a study of this size, any biomarker results can at best be considered hypothesis-generating. A biomarker observed prior to treatment at a higher rate in responders than in non-responders could be predictive or prognostic but might be hypothesized to be predictive if for example the marker were already known to be adverse – leading to a testable hypothesis in a future study. As no such data is examined in the manuscript, we have removed the reference to potential predictive biomarkers (see line 329 of the revised manuscript).

2.18 - (Specific comment 5) Line 277-278, was the half life reported correctly? i.e. the lowest dose has the longest half-life?

The median half-lives are reported correctly. We have added the 95% confidence intervals to the manuscript to illustrate that the differences are not significant (see lines 223ff of the revised manuscript).

RESPONSE TO REVIEWER #3

In this manuscript, the authors present data from phase 1/2 study on AML/HR MDS patients treated by an Axl inhibitor, bemcentinib, as single agent or in association with LDAC.

R/R and unfit AML are bad prognosis population with a high unmet medical need. AXL is a tyrosine kinase receptor with adhesion and environmental sensing capacities. Through its canonical ligand GAS6, it is activated by bypassing peripheral apoptotic bodies and the downstream pathways give it a role in cell survival and proliferation, as well as in migration and efferocytosis. Relatively ubiquitously expressed, Axl belongs to RTAMs, that generally play a major role in the immune system, particularly in the innate immune system. Axl plays also a major role in cancer resistance to treatment, in AML as in various others cancer, and many data support its association with other proapoptotic drugs to enhance their efficacy in a synergistic manner.

The topic is of interest, but the manuscript suffers from some shortcomings that prevent its publication in its current form.

MAJOR Results

3.1 - The patient disposition diagram should be provided with screen failure, figure should not be in supp. In contrast, for an early phase clinical trial, there are a lot of OS curves.

Authors would like to thank the reviewer for this comment and apologize for having accidentally omitted the numbers of screen failures. We have included them into the new Figure 1.

In addition, we have substantially reduced the number of survival plots shown and now focus on the analyses stratifying patients by whether they had been treatment-naive or R/R following advice of reviewer #1 (please see above, comment 1.18, and Figure 3 of the revised manuscript).

3.2 - Characteristics for patients should include cytogenetic risk in both Tables 1 and 2, molecular profile for most prevalent/important mutated genes (at least NPM1, FLT3, TP53) and also previous allo-HSCT and previous exposition to venetoclax or intensive chemotherapy for R/R cohort

Cytogenetic risk and prior therapy have been added to Table 1 (Please note that the original tables 1 and 2 have been combined into Table 1 based on another reviewer's comment). However, mutation status was only routinely recorded for FLT3.

3.3 - For fatal TEAEs, please precise if event occurred in patients with active disease or in CR

None of the patients with fatal TEAEs had a CR. We have added a statement to the Safety and Tolerability section (see line 165 of the revised manuscript).

3.4 - At the beginning of ASP2215/gilteritinib development, there was a warning about Axl inhibition and eyesight problems due to RTAM role that has been demonstrated in MERTK^{-/-} mice with retinitis pigmentosa, since the cells of the retinal epithelium can no longer ingest apoptotic bodies, inducing inflammation and retinal degeneration. Please provide a specific sentence in ophthalmological AE, even if there is not.

Authors would like to thank the reviewer for this insightful comment. We have included a sentence on the observed adverse events associated with the eyes in the Safety and Tolerability section (see lines 182ff of the revised manuscript), along with a brief section in the discussion (see lines 263ff of the revised manuscript).

However, we do not believe that retinal toxicity should be considered a likely side effect of treatment with bemcentinib, which is a selective AXL inhibitor.

There is abundant evidence that loss of MERTK, a member of the same subfamily of receptor tyrosine kinases as AXL, is associated with retinal dystrophies, ranging from rat (D'Cruz 2000) and mouse (Duncan 2003) models to identification of MERTK mutations in individuals and families with retinal dystrophies (Gal 2000, Thompson 2002, Eberman 2007, Mackay 2010, Ostegaard 2011, Ksantini 2012).

However, despite the existence of suitable mouse knockout models for AXL, no similar phenotype or association has been described for AXL. Although combined AXL/MERTK inhibitors can induce retinal toxicity in a mouse model, this toxicity was not observed with a closely related selective AXL inhibitor (Inoue 2021). Indeed, the only published link between AXL and retinopathies that we are aware of demonstrates that AXL activity (not loss) is important in diabetic retinopathy and that inhibition of AXL could be a candidate therapy for diabetic retinopathy (Wu 2021). The inhibitor mentioned, ASP2215/gilteritinib, has significant activity against multiple kinases including MERTK (Lee et al 2017) and this may account for any warnings/caution that were expressed.

- D'Cruz PM, Yasumura D, Weir J et al (2000) Mutation of the receptor tyrosine kinase gene MerTK in the retinal dystrophic RCS rat Hum Mol Genet 9,645-651
- Duncan JL, LaVail MM, Yasumura D et al. An RCS-like retinal dystrophy phenotype in mer knockout mice. Investigative Ophthalmology & Visual Science. 2003a;44:826–838.
- Ebermann I, Walger M, Scholl HPM et al. Truncating mutation of the DFNB59 gene causes cochlear hearing impairment and central vestibular dysfunction. Hum. Mutat. 28: 571-577, 2007
- Gal A, Li Y, Thompson DA et al. Mutations in MERTK, the human orthologue of the RCS rat retinal dystrophy gene, cause retinitis pigmentosa. Nature Genet. 26: 270-271, 2000
- Inoue S, Yamane Y, Tsukamoto S et al. Discovery of a potent and selective Axl inhibitor in preclinical model. Bioorg Med Chem. 2021;39:116137.
- Ksantini M, Lafont E, Bocquet B et al. Homozygous mutation in MERTK causes severe autosomal recessive retinitis pigmentosa. Europ. J. Ophthal. 22: 647-653, 2012.
- Lee LY, Hernandez D, Rajkhowa T, et al. Preclinical studies of gilteritinib, a next-generation FLT3 inhibitor. Blood. 2017;129(2):257-260.
- Mackay, DS, Henderson RH, Sergouniotis, PI et al. Novel mutations in MERTK associated with childhood onset rod-cone dystrophy. Molec. Vision 16: 369-377, 2010.
- Ostergaard E, Duno M, Batbayli M et al. A novel MERTK deletion is a common founder mutation in the Faroe Islands and is responsible for a high proportion of retinitis pigmentosa cases. Molec. Vision 17: 1485-1492, 2011
- Thompson, DA, McHenry CL, Li Y et al. Retinal dystrophy due to paternal isodisomy for chromosome 1 or chromosome 2, with homoallelism for mutations in RPE65 or MERTK, respectively. Am. J. Hum. Genet. 70: 224-229, 2002.
- Wu W, Xu H, Meng Z, et al. Axl Is Essential for *in-vitro* Angiogenesis Induced by Vitreous From Patients With Proliferative Diabetic Retinopathy. Front Med (Lausanne). 2021;8:787150.

3.5 - Treatment efficacy: % of patients bridged to transplant should be provided. Indeed, median OS in R/R disease for patient unfit for intensive chemotherapy and with relatively low CR/CRi rate are surprisingly long

No patients were bridged to transplant. The reason for this is that only patients unfit for intensive chemotherapy were included into the LDAC arm of the trial and thus no patients were transplant-eligible. We included a corresponding statement into the manuscript (see lines 213f of the revised manuscript). In the dose-escalation arm, patients were to have already undergone transplantation if eligible (see Supplementary Methods 1 of the revised manuscript).

3.6 – Please provide median time to ANC recovery, platelet recovery, time to CR etc.

We have added median time to CR in the Treatment Efficacy section (see lines 204ff of the revised manuscript). Time to ANC recovery and time to platelet recovery were not planned endpoints in this study and thus not uniformly recorded. However, median time between LDAC doses, which depended on adequate platelet and ANC recovery,

was 35 days. We included this information into the revised manuscript (see lines 191f of the revised manuscript).

MINOR

Title

3.7 – Minor: Intensive chemotherapy would be more informative for NC readers

The title has been updated accordingly.

Abstract

3.8 Minor: line 59, AML and HR MDS would be more precise

As noted also in our reply to comment 2.13 of reviewer #2, the manuscript originally described only the LDAC cohort, which exclusively focuses on AML. We regret that when the manuscript was updated per editorial request to include the dose escalation cohorts, which include two high-risk MDS patients, the abstract was not sufficiently updated. We have added text to clarify that there are both AML and HR MDS patients (see line 61 of the revised manuscript).

3.9 Minor: abbreviation w/o definition

We have introduced the abbreviation “LDAC” at the appropriate location (see line 62 of the revised manuscript).

3.10 Minor: last sentence about overall efficacy in “second line” should be replaced by objective data.

We have replaced this sentence with objective data (see lines 75f of the revised manuscript).

Introduction

3.11 Minor: line 88, ref (Burnet Cancer 2007, LDAC in unfit AML) is missing

We have added this reference (see line 91 of the revised manuscript).

3.12 Minor: several abbreviations, examples OS line 100 previously defined line 93

We have revised the manuscript text to remove previously defined abbreviations (see, for instance, line 102 of the revised manuscript).

Methods

3.13 Minor: line 386, reasons for unsuitability for intensive chemotherapy should be defined

Reasons for unsuitability for intensive chemotherapy included advanced age (defined as >75 years). Altogether, the overall assessment of comorbidities leading to unfit for intensive chemotherapy was left at the discretion of the treating physician. The protocol defines some of the most relevant comorbidities in the inclusion / exclusion criteria (included in Supplementary Methods 1). These include abnormal organ function (renal, liver, cardiac) and ECOG PS >2. We have added more details to the Patients section in order to clarify this issue (see lines 332ff of the revised manuscript).

3.14 Minor: line 394, explain why exclusion of prior treatment of gilteritinib, all NC readers doesn't know that it is a dual FLT3/AXL inhibitor

We would like to thank the reviewer for pointing out this issue. We have added the corresponding explanation (see lines 344ff of the revised manuscript).

3.15 Minor: Study treatment line 410, please provide an administration schedule. Is there a rest between D21 bemcentinib and D28 LDAC? or bemcentinib is given continuously in D21-cycle and LDAC for 10 days in D28-cycles?

Bemcentinib is dosed once daily during continuous 21-day treatment cycles, without rest periods and LDAC for 10 days in 28-day cycles. However, LDAC administration could be delayed depending on persisting myelosuppression. We included a corresponding statement into our manuscript (see lines 370ff of the revised manuscript).

3.16 Minor: Efficacy assessment: Ref is missing for IWG and author should also discuss why they do not use ELN response criteria for AML that represent 94% of the cohort. CR and CRi are presented in Table and should be detailed here, same thing for ORR, DCR.

The earlier versions of the protocol specified IWG as response criteria also for AML because the ELN response criteria for AML were published after the study started in 2014. However, the ELN criteria were adopted in a revision to the protocol and the patients were therefore actually assessed according to ELN criteria. We have added references for the response criteria and summarised the different responses with their associated criteria. To reduce the length of the final manuscript, we have moved these details into Supplementary Methods 5.

3.17 Minor: Please provide the way to calculate QTc (Fridericia ?)

We have added the formula and reference in a new Supplementary Methods section (Supplementary Methods 2) that also summarises the different grades of QTc prolongation.

Results

3.18 Minor: Cohort A does not refer to anything clearly defined at this stage, "dose escalation cohort" should be used

We have implemented the suggested replacement throughout the text (see, for instance, line 121 of the revised manuscript).

3.19 Minor: Cytogenetic risk in Table 2 should be defined (Grimwade, Blood 2010?) and paper should be referenced line 155 or in Methods

Cytogenetic risk categories were reported by the individual sites using their own local criteria. Where detailed karyotypes were provided, they are consistent with the

categories in the ELN 2017 guidelines (Döhner 2017). We have added a footnote to Table 1 (see lines 678ff of the revised manuscript).

3.20 Minor: for all tables, abbreviations are missing

The tables now include footnotes explaining the abbreviations.

3.21 Minor: line 164, the introduction “bemcentinib as monotherapy was generally weel-tolerated” is a conclusion. The sentence should be deleted.

We have removed the sentence from our manuscript (see line 144 of the revised manuscript).

3.22 Minor: line 193, a “.” is surplus.

We rewrote the Safety and Tolerability section (see lines 145ff of the revised manuscript) based on other reviewer suggestions, thereby removing this spelling mistake.

3.23 Minor: line 219, the introduction “bemcentinib in combination with LDAC was generally weel-tolerated” is a conclusion. The sentence should be deleted.

We have removed the sentence from the manuscript. The revised Safety and Tolerability section (see lines 145ff of the revised manuscript) no longer includes it.

REVIEWER COMMENTS

Reviewer #1 (Remarks to the Author):

Thank you for revising the manuscript based on my comments.

I just have several minor comments that would improve the manuscript.

1.1 In the Abstract- I would recommend clarifying that dose escalation was done as a single agent in R/R AML and that there was an expansion cohort of combination therapy with LDAC in both newly diagnosed and R/R AML. It is confusing that clinical activity is reported in both newly diagnosed and R/R AML in the abstract without further clarification about eligibility and study design.

Authors would like to thank the reviewer for pointing this out. We have revised the text to clarify this. Please see lines 61ff of the revised manuscript.

Please note that there were two treatment-naïve patients and two MDS patients in the dose escalation cohort so we have not restricted our wording to R/R AML.

1.2 I would also revise the last statement in the Abstract and not state a "positive signal" was observed and just state that these results are promising and warrant further investigation.

We changed the abstract according to the reviewer`s suggestion. Please see line 77 of the revised manuscript.

Reviewer #2 (Remarks to the Author):

Most of my previous comments were addressed to my satisfaction. However, there are still a few issues that deserve authors' attention.

2.1. The sample size assumption is the same for B2 and B5 but the actual sample size are different. I understand that the cohort is exploratory but please provide a rationale why the sample sizes are different.

Authors would like to thank the reviewer for raising this issue. We observed promising efficacy activity of the combination bemcentinib + LDAC in part B2 (n=14). However, the patient population included newly diagnosed and R/R patients and was thus too heterogenous for meaningful conclusions. Therefore, part B5 was added to investigate n=14 patients with relapsed and refractory AML each. However, following regulatory feedback from the FDA, the cohort was reduced to 20 patients combined, with no more than 6 refractory, ensuring at least 14 relapsed patients. Thus, the sample size calculations were followed for the relapsed patients while the number of refractory patients was reduced to meet the advice from the FDA. We have added this information (Please see lines 333ff of the revised manuscript).

2.2. Line 316. What is “a test of size 0.2”? Do you mean a “one-sided type I error of 0.2”?

Authors would like to apologize for this error. We have rephrased this section (also considering your comment 2.1) to clarify the sample size calculations (Please see lines 333ff of the revised manuscript).

2.3. Line 326-329. All endpoints should be clearly defined. This was pointed out previously but was not updated sufficiently.

2.3.1.1. For ORR and DCR, the patients who are evaluable for the endpoint (i.e. the denominator) and the definition of success (i.e. the numerator) should be detailed.

2.3.1.2. For RFS, EFS, OS, the event/censor should be defined. For example, what is the “events” considered for event-free survival? What is the start time, registration or first dose?

We apologise for misunderstanding the level of detail requested. The requested information was added to Supplementary Methods 5 and readers are pointed to this information in line 351 and lines 434f of the revised manuscript. Furthermore, we have now added the long versions of all abbreviations corresponding to our endpoints. Please see lines 350f of the revised manuscript.

4. Figure 1, things do not add up. 166 patients were screen and 43 were excluded which implies 123 were enrolled but the manuscript text states 122 were enrolled.

We apologise for this mistake. Two patients decided not to join the study (not 1, as originally stated). Thus, the correct number of excluded patients is 44. We corrected this in Figure 1 of the revised manuscript.

5. Figure 1, the right-hand side box of “phase2a, other cohorts”. Should the cohorts listed be B4 instead of B5?

Thank you for spotting this. You are correct that this should have read B4 instead of B5. This has been corrected in Figure 1 of the revised manuscript.

Reviewer #3 (Remarks to the Author):

All my concerns have been addressed in the revisions.

We are very happy that we were able to address all your concerns. Many thanks again for your invaluable help with strengthening our manuscript.